# AutoRPA: Efficient GUI Automation through LLM-Driven Code Synthesis from Interactions

**Minghao Chen** [1]   **Xinyi Hu** [1]   **Zhou Yu** [1]   **Yufei Yin** [1]

## Abstract

Large Language Model (LLM) based agents have demonstrated proficiency in multi-step interactions with graphical user interfaces (GUIs). While most research focuses on improving single-task performance, practical scenarios often involve repetitive GUI tasks for which invoking LLM reasoning repeatedly, i.e., the ReAct paradigm, is inefficient. Prior to LLMs, traditional Robotic Process Automation (RPA) offers runtime efficiency but demands significant manual effort to develop and maintain. To bridge this gap, we propose **AutoRPA**, a framework that automatically distills the decision logic of ReAct-style agents into robust RPA functions. AutoRPA introduces two core innovations: (1) A *translator-builder pipeline* where a translator agent converts hard-coded ReAct actions into soft-coded procedures, and a builder agent synthesizes robust RPA functions via retrieval-augmented generation over multiple trajectories; (2) A *hybrid repair strategy* during code verification, combining RPA execution with ReAct-based fallback for iterative refinement. Experiments across multiple GUI environments demonstrate that RPA functions generated by AutoRPA successfully solve similar tasks while reducing token usage by 82%~96%, significantly improving runtime efficiency and reusability.

## 1. Introduction

Graphical User Interfaces (GUIs) are the primary medium for human interaction with computers and mobile devices. Recently, Large Language Model (LLM) agents (Hong et al., 2024; Xi et al., 2025) have demonstrated remarkable versatility in handling various GUI tasks. Most current research focuses on improving the performance of LLM on novel, individual tasks, such as by designing more sophisticated workflows (Wang et al., 2024c;b) or by introducing end-to-end reinforcement learning to enhance reasoning capabilities (Wang et al., 2025a). However, in practical deployment scenarios, a substantial portion of GUI tasks are *repetitive in nature*: the same user may perform identical activity daily (e.g., filing reports), or different users may share common needs (e.g., booking flights). For such recurring task types, rather than invoking expensive LLM reasoning for each instance, a more desirable solution is to generate *reusable, low-cost automation functions* that can be efficiently executed across similar tasks.

Prior to LLMs, traditional Robotic Process Automation (RPA) (Ivančić et al., 2019) addressed repetitive GUI tasks by executing manually crafted scripts that simulate user actions such as clicking and typing. While efficient at runtime, RPA systems require significant human effort to develop and maintain, and are brittle to GUI layout changes (Hellmann & Maurer, 2011; Qian et al., 2020). Therefore, **the aim of this paper** is to investigate how to automatically synthesize token-efficient RPA functions for a specified task type with the assistance of LLMs. Crucially, the generated RPA code should handle variations in both the environment and task instructions.

One direct solution is to have the LLM generate executable code directly based on the task description (Kim et al., 2024). This method can be grounded in the Plan-and-Execute paradigm (Kim et al., 2024; Sun et al., 2023), where the LLM agent generates the plan or code and then subprocesses execute it. However, LLMs often struggle to produce robust, long-horizon GUI code without curated expert knowledge of the environment (Valmeekam et al., 2022). An alternative is skill learning (Chen et al., 2024; Qian et al., 2024; Zhao et al., 2024), which stores the success trajectory of the LLM agent as exemplars for subsequent similar tasks. However, these methods often struggled with generalizing skills to cope with different scenarios (Chen et al., 2024), and relied heavily on the LLM's intrinsic abilities to handle new tasks.

[1]Zhejiang Key Laboratory of Space Information Sensing and Transmission, School of Computer Science, Hangzhou Dianzi University, China. Correspondence to: Zhou Yu <yuz@hdu.edu.cn>.

*Proceedings of the 43ʳᵈ International Conference on Machine Learning*, Seoul, South Korea. PMLR 306, 2026. Copyright 2026 by the author(s).

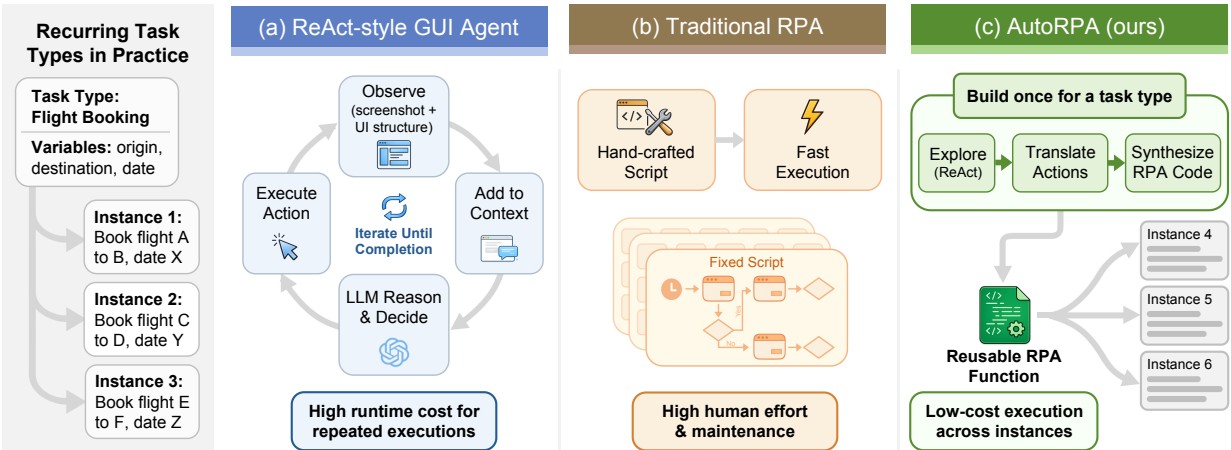

*Figure 1.* Comparison of GUI automation paradigms. (a) ReAct-style LLM agents achieve high flexibility but incur substantial per-instance costs, unsuitable for repetitive tasks. (b) Traditional RPA offers efficiency for repetitive tasks but requires manual scripting. (c) AutoRPA automatically synthesizes robust, low-cost RPA functions for arbitrary task types from LLM agent interactions.

To this end, we propose **AutoRPA**, a general framework that utilizes LLM agents to generate high-quality RPA code for GUIs. Our framework "distills" the decision logic of ReAct agents into executable code: AutoRPA first utilizes a ReAct-style agent to explore and collect successful trajectories. To facilitate RPA code synthesis, we introduce a translator agent that transforms each hard-coded ReAct step into a soft-coded action, enabling dynamic execution in varying environments. A builder agent then synthesizes robust RPA functions from these soft-coded trajectories, incorporating necessary logic and conditionals.

Moreover, the completion process of different specific instructions may vary in different situations. Therefore, to derive a robust RPA code, the builder is augmented with a retrieval mechanism that indexes a tree-structured trajectory database, enabling access to multiple past trajectories. During code verification, rather than simply refining and retrying, we employ a hybrid repair strategy: if the execution of synthesized code fails, the ReAct agent resumes execution from the breakpoint, producing a corrective demonstration that the builder can incorporate for refinement.

Our main contributions are as follows:

- We introduce **AutoRPA** to address a practical problem: generating robust and efficient RPA functions for task types, targeting repetitive GUI automation scenarios.
- We propose a **translator-builder pipeline** that transforms stepwise ReAct actions into soft-coded actions, and synthesizes robust RPA functions via a tree-structured trajectory retrieval.
- We develop a **hybrid repair strategy** that combines direct code verification with ReAct-based fallback, enabling iterative improvement of synthesized code.
- Experiments on three GUI benchmarks demonstrate that AutoRPA can perform token- and time-efficient GUI au-

tomation. Especially, AutoRPA reduces token usage by up to 96% while maintaining or exceeding the success rate of LLM-based GUI agents.

## 2. Related Works

### 2.1. GUI Automation via Traditional Methods

Robotic Process Automation (RPA) (Ivančić et al., 2019) has long been used to enhance productivity by automating repetitive tasks within Graphical User Interfaces (GUIs). Traditional RPA systems automate routine tasks by mimicking human actions, primarily relying on manually crafted scripts or predefined rules (Hellmann & Maurer, 2011; Qian et al., 2020). However, developing RPA demands significant domain expertise and requires frequent manual updates to accommodate changes in GUI layouts and task requirements, thereby limiting its scalability and flexibility in diverse and dynamic scenarios (Zhang et al., 2024). To overcome the limitations of traditional RPA, subsequent research has explored learning-based approaches (Humphreys et al., 2022; Liu et al., 2018). These methods train deep learning models to predict and execute actions based on GUI observations. While demonstrating greater potential, these learning-based techniques often necessitate substantial amounts of GUI interaction data for training and still face challenges regarding stability and generalization when deployed in real-world applications.

### 2.2. LLM Agents for GUI Automation

Within the field of GUI automation, LLM agents leverage the emergent reasoning capabilities of large language models (OpenAI, 2023; Ouyang et al., 2022; Wei et al., 2022) and have been developed for applications across web, mobile, and desktop platforms. GUI agents such as See-

Act (Zheng et al., 2024) and WebVoyager (He et al., 2024) leverage the multimodal capabilities of GPT-4V (OpenAI, 2023), which interact with dynamic web pages by processing both screenshots and HTML inputs. AppAgent (Zhang et al., 2025) also utilizes GPT-4V to handle screenshots and XML data, allowing for the completion of various tasks within mobile applications.

Beyond basic interaction, advanced studies aim to improve LLM agents through experience-based optimization. Reflexion (Shinn et al., 2023) enables the LLM agent to reflect on failed trajectories and revise its behavior in future attempts. ICE (Qian et al., 2024) and ExpeL (Zhao et al., 2024) build a skill library from successful execution traces to guide subsequent similar tasks. AutoManual (Chen et al., 2024) and Mobile-Agent-E (Wang et al., 2025b) further allow the LLM to induce environmental rules from interactions. Compared to our methods, these approaches focus on the self-evolution of LLM agents, and the decision-making process still relies on the LLM during testing.

## 2.3. LLMs for Code Generation

LLMs have demonstrated impressive performance in natural-language-to-code (NL2Code) tasks (Jiang et al., 2026; Du et al., 2024), rivaling expert human programmers on benchmarks like HumanEval (Du et al., 2024) and MBPP (Austin et al., 2021). However, in novel or intricate environments like GUIs or games, generating the complete planning code upfront poses significant challenges for LLMs (Valmeekam et al., 2022; Sun et al., 2023). As a result, for these agentic environments, enabling LLMs to generate reliable code often requires thorough domain-specific knowledge and examples (Wang et al., 2024a). In contrast, our framework utilizes the iterative, step-by-step paradigm of ReAct agents (Yao et al., 2023) to explore and collect successful trajectories and then distills these into verified RPA code via translation and synthesis, without requiring curated knowledge or examples.

## 3. Methods

**Overview.** We propose the AutoRPA framework to build RPA code tailored to user-specified GUI task types. We first formulate the problem in Section 3.1. AutoRPA comprises three main phases, as illustrated in Fig. 2. **Exploration phase:** A ReAct-style agent iteratively interacts with the GUI environment, while a translator agent converts its hard-coded actions into soft-coded actions, detailed in Section 3.2. **RPA Generation phase:** Based on the translated trajectory, a builder agent synthesizes the RPA code, while actively retrieving relevant interaction information from a trajectory database, detailed in Section 3.3. **RPA Verification phase:** The newly generated code will be validated on the seen tasks. For the failed execution, an ana-

lyzer agent will analyze the breakpoint and use the ReAct agent to continue exploring for repairs, followed by code refinement, detailed in Section 3.4. Finally, we evaluate the obtained RPA code at the testing stage.

## 3.1. Problem Formulation

Formally, we model the GUI environment as a Partially Observable Markov Decision Process (POMDP): $(\mathcal{S}, \mathcal{A}, \mathcal{T}, \mathcal{G}, \mathcal{O})$. The task set $\mathcal{G}$ comprises totally $K$ task types: $\mathcal{G} = \bigcup_{k=1}^{K} \mathcal{G}^k$. At the start of each episode, the environment samples a task $g \in \mathcal{G}^k$ described in natural language from one of the task types, and initializes a scene $s_0 \in \mathcal{S}$. The environment will provide an initial observation $o_0 \in \mathcal{O}$, which is typically a screenshot of the current interface and textual information derived from the Document Object Model (DOM) or accessibility tree. The agent can interact with the GUI environment through the permissible action set $\mathcal{A}$ (e.g., click, type). After executing an action $a_t \in \mathcal{A}$ at time step $t$, the environment transitions to a new state $s_{t+1}$ according to the dynamics $T(s_{t+1}|s_t, a_t)$ and subsequently emits a new observation $o_{t+1} \sim O(o_{t+1}|s_{t+1})$. At the end of the episode, a binary reward $r(\tau) \in \{0, 1\}$ is assigned based on the final trajectory $\tau = (g, o_0, a_0, o_1, ..., a_T, o_{T+1})$, indicating whether the task $g$ was completed or not.

Our objective is to generate an RPA function $F_k$ for a user-specified task type $\mathcal{G}^k$. For any task instance $g \in \mathcal{G}^k$, this function $F_k$ should produce an action trajectory $\tau = F_k(g^k)$ that successfully completes the task, i.e., $r(\tau) = 1$. Given that the function $F_k$ might internally invoke LLMs to handle complex scenarios, we aim to construct and optimize $F_k$ to minimize the token consumption while guaranteeing task success. This objective is formulated as:

$$\min_{F_k} \mathbb{E}_{g\sim\mathcal{G}^k, s_0\sim\mathcal{S}}[1 - r(F_k(g)) + \lambda \cdot \text{cost}(F_k(g))], \quad (1)$$

where $\text{cost}(F_k(g))$ represents the token cost incurred during the process of $F_k$ for task $g$; $\lambda$ is the tradeoff between reward and cost. The prior approaches following the ReAct paradigm (Yao et al., 2023) typically perform step-by-step inference within $F_k(g)$, leading to substantial token costs. In contrast, existing methods that attempt to directly generate the entire code for $F_k(g)$ upfront (Sun et al., 2023) often struggle to consistently guarantee the success of the task. In our AutoRPA, we sample $N$ tasks $g_n \in \mathcal{G}^k, 1 \le n \le N$ to build a code style $F_k$ and address both limitations by combining ReAct exploration with iterative code synthesis and verification.

## 3.2. Exploration Phase

Starting with the first building task $g_1$, we generate its ReAct trajectory, and then save it into a trajectory bank. This section details the process of generating trajectories follow-

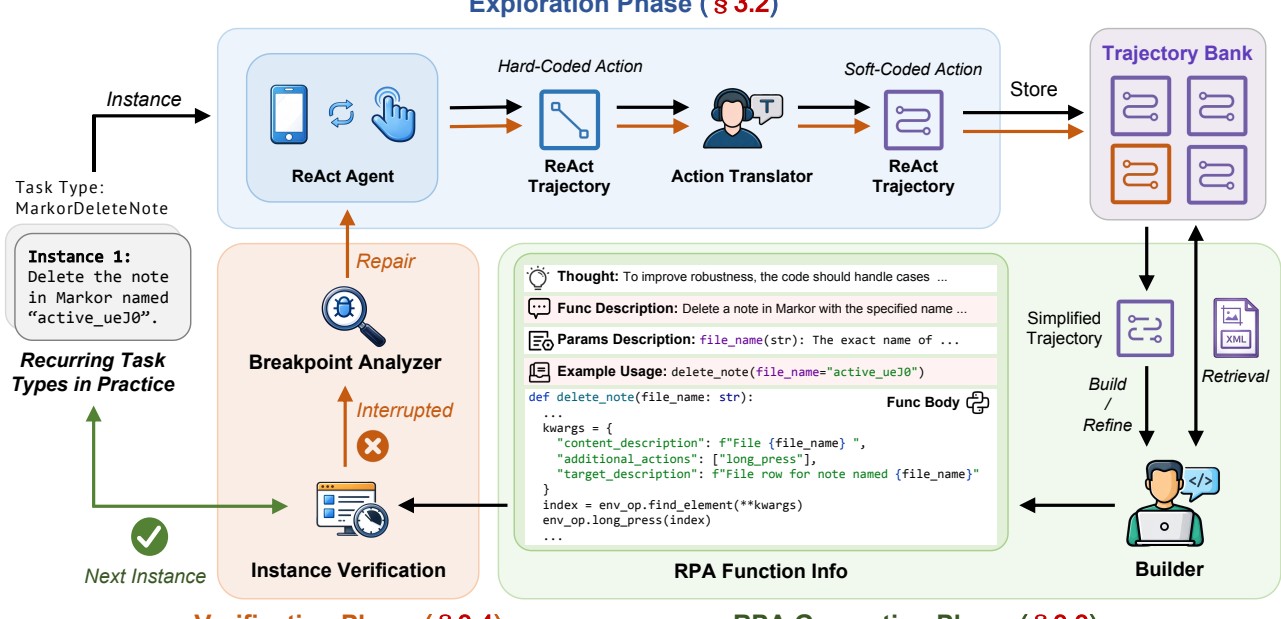

*Figure 2.* **AutoRPA Overview:** For a task in the target task type, AutoRPA explores and repairs bugs using the ReAct agent, while a translator agent converts the resulting actions into soft-coded actions. A builder then generates the RPA function based on the simplified trajectory from the trajectory bank. The newly generated code will be verified on the seen tasks. If it fails, it will be analyzed and repaired; if successful, the new task will be examined.

ing the ReAct paradigm and subsequently translating hard-coded ReAct actions into reusable soft-coded equivalents.

**ReAct Trajectory.** Our ReAct agent follows the implementations in prior works (Putta et al., 2024; Rawles et al., 2025). At each time step $t$, the ReAct agent receives the task instruction $g$, history of its outputs (including reasoning and actions), the execution summaries of previous actions ($\rho_0, ..., \rho_{t-1}$), and the current multimodal observation $o_t$, which consists of a screenshot annotated using the Set-of-Mark (SoM) technique (Yang et al., 2023) and a list of GUI elements extracted from the DOM or accessibility tree. It outputs: **(1) Analysis** of the current observation $o_t$; **(2) Checklist** of subtasks, indicating the steps completed and planned towards the goal; **(3) Next action** $a_t \in \mathcal{A}$ represented in JSON format, along with the reasoning for that action. Following M3A (Rawles et al., 2025), a separate summarizer agent is employed to summarize the effect of each executed action $a_t$ based on the multimodal observations recorded before and after execution, yielding $\rho_t$. At the end of the episode, a concluder agent generates a conclusion $C$ based on the action history and the environmental reward $r$. Therefore, a complete ReAct trajectory for the task $g$ can be represented as:

$$\tau_{\text{ReAct}}(g) = (g, o_0, a_0, \rho_0, o_1, ..., a_T, \rho_T, o_{T+1}, C). \quad (2)$$

To improve the success rate of the ReAct agent during the building stage, we adopt the reflection mechanism (Chen et al., 2024; Shinn et al., 2023). For a failed episode, the conclusion $C$ will also include a reflection, and the ReAct agent retries based on the reflection (up to $N_{\text{ref}}$ times).

Noticeably, the specific ReAct agent used here is modular and can be substituted with other capable GUI agents, such as Computer-Using Agent (CUA) from Anthropic (Anthropic, 2024) or OpenAI (OpenAI, 2024), as long as they can select the next action based on the GUI observation. These GUI agents typically specify actions using hard-coded identifiers (e.g., `click(index=2)` or `click(position=(144.5, 278.0))`) for versatility and performance (Zheng et al., 2024). However, these actions lack robustness, as they may fail if the GUI layout changes even slightly. To address this, we introduce a translator agent that converts hard-coded actions into soft-coded equivalents using element attributes.

**Translator Agent.** The translator agent processes each effective action (i.e., actions causing screen changes) to generate robust, reusable procedures for the subsequent RPA building. The inputs to the translator agent are similar to the summarizer agent: the output of the ReAct agent at step $t$ and the before-and-after multimodal observations. The translator then outputs: **(1) Robustness analysis**: Evaluates the environmental dependency and failure modes of action $a_t$. **(2) Action translation**: Converts the hard-coded action $a_t$ into a soft-coded equivalent $a'_t$. For example:

```python
# Find the target element using its properties
target_element = env_op.find_element(text="password",
    editable=True, target_description="Input field
    for password at the center of the screen")
assert target_element != -1, "Cannot find password
    input field."
# Perform the action on the located element
env_op.input_text(target_element, "123456")
```
Python

This translation involves generating a Python code snippet that: (1) locates the target GUI element based on its semantic attributes (e.g., 'text content', 'element type', 'available actions') rather than its index or absolute position. We provide an additional **find_element** function that can locate target elements based on matching input attribute values. Furthermore, for accuracy in positioning, when multiple elements that meet the criteria are found, the GUI grounding model (e.g., GPT-4o) will be employed for localization based on the 'target_description'; (2) optionally includes assertion statements within the code to verify that the action achieves the expected outcome in the subsequent observation, as well as fallback operations.

This translation enables dynamic element identification while maintaining operational equivalence, significantly improving action trajectory portability across interface variations and easing the subsequent RPA building. Finally, we get the translated ReAct trajectory:

$$\tau'_{\text{ReAct}}(g) = (g, o_0, a'_0, \rho_0, o_1, ..., a'_T, \rho_T, o_{T+1}, C). \quad (3)$$

### 3.3. RPA Generation Phase with RAG

**Builder Agent.** After completing a building task $g_1$ with the ReAct agent, we employ a builder agent to generate a robust RPA function $F_k$ for task type $\mathcal{G}^k$. The input for the builder agent includes: the task text $g_1$, the variables in the task text $g_1$ (specified by the user or determined by the builder), and the translated ReAct trajectory of the task. Since the complete ReAct trajectory $\tau'_{\text{ReAct}}(g)$ contains excessively long observational information, we only provide the simplified version, i.e., $\psi(\tau'_{\text{ReAct}}(g)) = (g, a'_0, \rho_0, .., a'_T, \rho_T, C)$, with the original observations removed, while the complete version $\tau'_{\text{ReAct}}(g)$ is stored in a trajectory bank $\mathcal{D}_\tau$.

For the refinement of previously generated code, the input of the builder agent will additionally include: (1) the current function code $F_k$, (2) its execution and repair trajectory $\tau'_{\text{hybrid}}(F_k, g_*)$ (described in Section 3.4), (3) conclusions $C$ from the trajectories of prior explorations or repairs. Considering that the code may have been refined multiple times, we only provide the simplified trajectory of the last execution for brevity.

**Builder with RAG.** However, we found that when exact observations were not provided to the builder agent, it occa-

sionally generated erroneous assumptions about the actual interface state or misunderstood the rationale behind the executed actions, leading to redundant or inefficient code. To address this, we adopt the Retrieval-Augmented Generation (RAG) mechanism (Gao et al., 2023) to allow the builder agent to proactively retrieve relevant observations from the database $\mathcal{D}_\tau$. Specifically, we adapted a Tree-organized Retrieval paradigm (Chen et al., 2023; Sarthi et al., 2024), treating each interaction step $(o_t, a'_t, \rho_t, o_{t+1})$ as a document block. Notably, $o_{t+1}$ is the overlapping content in two adjacent document blocks. Our **tree-organized database structure** is as follows:

1. The bottom layer preserves each interaction block $(o_t, a'_t, \rho_t, o_{t+1})$ of the trajectory.
2. The middle layer contains simplified trajectory $\psi(\tau'_{\text{ReAct}}(g))$ where action results $\rho_t$ serves as a summary of the corresponding interaction block.
3. The top layer stores the final conclusion $C$, which serves as the summary of each trajectory.

Based on the tree-organized database, we provide the builder agent with a tool function: **fetch_info(traj, step=None)**. When invoked with a trajectory identifier and (optional) time step $t$, this function returns details from a lower layer: simplified trajectory $\psi(\tau'_{\text{ReAct}}(g))$ or multimodal interaction block $(o_t, a'_t, \rho_t, o_{t+1})$.

The output process of the builder agent begins by describing its own reasoning process. If it determines that more information is needed for coding, it can iteratively call the **fetch_info** tool until no further context is required. The builder agent then generates the RPA function $F_k$, which includes the function name and its description, input parameter description, function body, and function usage example, as shown in Fig. 3. The generated code prunes redundant steps from the translated ReAct trajectory and incorporates logical statements (conditions and loops) to enhance robustness against task variations and environmental changes. Additionally, considering that certain page information can only be extracted using MLLM, the builder agent can, following Adaplanner (Sun et al., 2023), include a special action function `env_op.ask_mllm(question, respond_format)` in the code.

### 3.4. Verification Phase with Hybrid Repair

**RPA Execution.** During validation and testing, it will be checked whether there is an RPA function for the current task. If available, the LLM needs to fill in the appropriate function parameters based on the task instruction and the RPA function, and then execute.

**Hybrid Repair.** After the builder agent generates a new RPA function $F_k$, we validate it through a rigorous process. We execute $F_k$ on the seen tasks $\{g_1, ..., g_n\}$ until encountering the first failure task $g_*$. Rather than directly request-

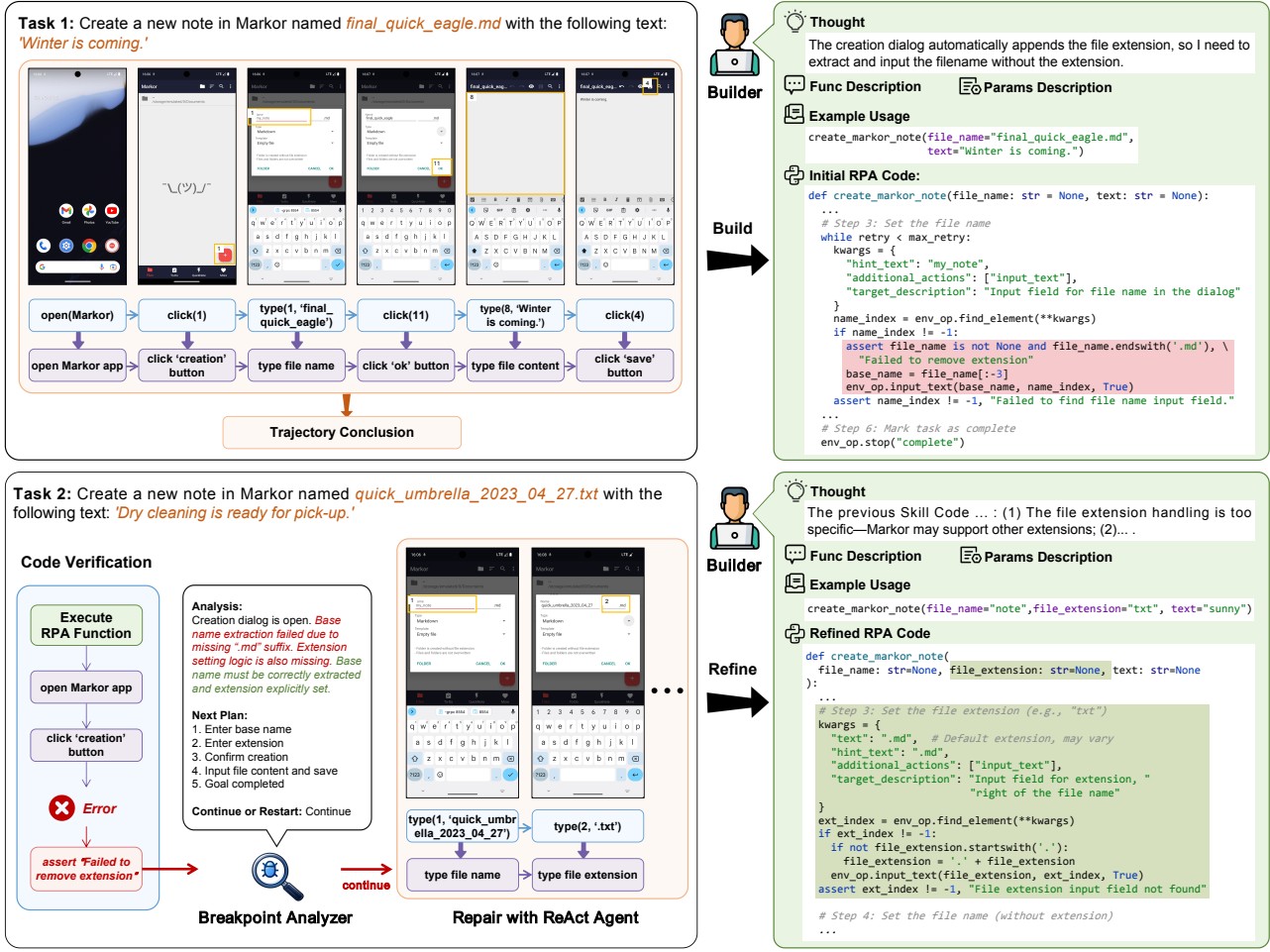

*Figure 3.* **Illustration of Code Generation and Refinement:** Based on the initial exploration trajectory, the generated RPA code failed to generalize across scenarios. Through the verification and hybrid repair processes, the builder agent can improve the robustness of the RPA code.

ing the builder to debug, we introduce an **analyzer agent** to analyze the breakpoint, as shown in Fig. 3. This agent needs to analyze the action trajectory that the code has executed, i.e., $F_k(g_*)$, and the observation $o_{t*}$ at the breakpoint, subsequently producing the failure cause, completed subtasks, the feasible continuation plans, and whether the task can be completed at the breakpoint or needs to be restarted. Based on the judgment of the analyzer, the ReAct agent then proceeds to complete the task $g_*$ (as described in Section 3.2), either resuming from the breakpoint or restarting from the initial state $s_0$ (and $t*$ will be set to 0). In the end, the hybrid trajectory $\tau'_{\text{hybrid}}$ that combines the trajectory of code execution and the ReAct trajectory will be obtained:

$$\tau'_{\text{hybrid}}(F_k, g_*) = F_k(g_*) \oplus (A, o_{t*}, a'_{t*}, \rho_{t*}, ..., o_{T+1}, C),$$
$$\psi(\tau'_{\text{hybrid}}(F_k, g_*)) = \psi(F_k(g_*)) \oplus (A, a'_{t*}, \rho_{t*}, ..., C),$$

where $\oplus$ is the concatenation operation and $A$ is the analyzer output. We then provide the simplified hybrid trajec-

tory $\psi(\tau'_{\text{hybrid}})$ to the builder agent for refinement as described in Section 3.3.

If the current RPA code has passed all the seen tasks, the next task $g_{n+1}$ will be taken from the building tasks for verification. For each task, we allow the builder agent to modify the RPA code $M$ times to pass the validations on all seen tasks. However, if the code still cannot pass all validations after $M$ modifications, we consider it difficult to automate $\mathcal{G}^k$ with RPA code. Therefore, we adopt the ReAct agent to complete these tasks during testing. For comparison, we also evaluate a variant of AutoRPA that disables ReAct during testing, referred to as **AutoRPA (code only)**.

## 4. Experiments

**Benchmarks.** To facilitate experiments on repetitive tasks, we need that the task type in the benchmark can be instantiated into multiple specific tasks. Therefore, we conduct the experiments on these three GUI environments: (1) **An-**

*Table 1.* Testing success rate, average execution time, and token consumption of LLM agent methods on AndroidWorld.

| Method | Model | Time (min) ↓ | Tokens (k) ↓ | Success (%) ↑ |
|---|---|---|---|---|
| SeeAct (Zheng et al., 2024) | GPT-4o | 6.38 | 68.4 | 15.3 |
| M3A (Rawles et al., 2025) | GPT-4o | 2.83 | 120.2 | 34.5 |
| ReAct[†] | GPT-4o | 4.97 | 79.9 | 35.2 |
| **AutoRPA (code only)** | GPT-4o | **1.08** | **2.6** | 34.3 |
| **AutoRPA** | GPT-4o | 1.76 | 14.7 | **37.0** |
| SeeAct (Zheng et al., 2024) | GPT-4.1 | 5.14 | 58.8 | 25.4 |
| M3A (Rawles et al., 2025) | GPT-4.1 | 2.23 | 103.4 | 48.3 |
| ReAct[†] | GPT-4.1 | 3.91 | 68.7 | 50.0 |
| **AutoRPA (code only)** | GPT-4.1 | **1.42** | **2.7** | 47.2 |
| **AutoRPA** | GPT-4.1 | 1.81 | 12.8 | **51.7** |
| SeeAct (Zheng et al., 2024) | GPT-5 | 12.43 | 134.5 | 40.3 |
| M3A (Rawles et al., 2025) | GPT-5 | 5.56 | 164.6 | 57.0 |
| ReAct[†] | GPT-5 | 8.57 | 142.5 | 74.1 |
| **AutoRPA (code only)** | GPT-5 | **2.72** | **6.2** | 70.7 |
| **AutoRPA** | GPT-5 | 4.35 | 30.6 | **75.9** |

**droidWorld** (Rawles et al., 2025) is a realistic Android environment featuring 116 task types across 20 real-world Android applications. This characteristic significantly supports our code generalization testing. AndroidWorld provides the screenshot and the Android accessibility tree for observations. The action space mimics human interactions, including operations such as tapping, long-pressing, and swiping. (2) **WebArena** (Zhou et al., 2024): A realistic web automation benchmark featuring self-hosted websites that simulate real-world applications. Our experiments focus on the Reddit domain containing 19 task types. (3) **MiniWoB++** (Liu et al., 2018) is a simulated web environment where agents perform various tasks on websites using keyboard and mouse operations. Following prior research (Chen et al., 2024; Kim et al., 2023; Sun et al., 2023), we selected a subset of MiniWoB++ tasks comprising 9 task types with environmental feedback and 44 types without feedback. We designate the 9 types with feedback as 'hard' tasks and the remaining 44 as 'simple' tasks.

**Compared Methods.** On AndroidWorld and MiniWoB++, we compare AutoRPA with these LLM-based GUI agents: **ReAct[†]** (Yao et al., 2023) denotes the ReAct agent as implemented in Section 3.2. **SeeAct** (Zheng et al., 2024), **M3A** (Rawles et al., 2025) and **SteP** (Sodhi et al., 2024) also follow the ReAct paradigm but with different implementations. **RCI** (Kim et al., 2023) is ground in Plan-and-Execute paradigm. **AdaPlanner** (Sun et al., 2023) belongs to the skill learning paradigm of Plan-and-Execute agents. **AutoGuide** (Fu et al., 2024) and **AutoManual** (Chen et al., 2024) belong to the skill learning paradigm of ReAct agents and additionally optimize a set of environmental rules.

**Implementation Details.** We employ GPT-4o, GPT-4.1, or

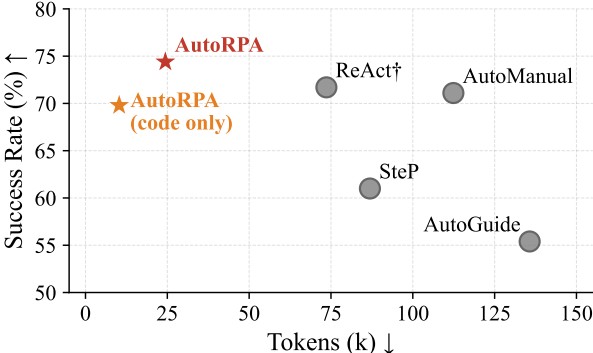

*Figure 4.* Testing success rate and token consumption of different methods with GPT-5 on WebArena (Reddit).

GPT-5 as the LLM backbone of our agents. For Android-World, we additionally evaluate with Claude-4.5-sonnet as the backbone to demonstrate that our method can benefit from better backbones (results are provided in the Appendix). During the building stage, we sample $N=3$ tasks for each task type, the ReAct agent can reflect and retry up to $N_{\text{ref}}=2$ rounds, and the builder agent is allowed to regenerate the code $M=3$ times to pass all validations.

### 4.1. Main Results

**Main Results on AndroidWorld.** As shown in Tab. 1, AutoRPA outperforms the existing ReAct-style agents while consuming less than half of the time and 18.6% of the tokens that ReAct[†] requires. One might wonder: *If AutoRPA uses the trajectories of ReAct to build RPA, why is the success rate of the tests higher than that of ReAct itself?* The reason is that AutoRPA uses ReAct with Reflection during the building stage, which ensures successful trajecto-

*Table 2.* Testing success rate and token consumption of LLM agent methods on 9 hard and all 53 task types in MiniWoB++ with GPT-4.1. "AdaPlanner (one demo)" restricts human examples to one.

| Methods | Hard (9 types) | | All (53 types) | |
|---|---|---|---|---|
| | Tokens (*k*) ↓ | Success (%) ↑ | Tokens (*k*) ↓ | Success (%) ↑ |
| RCI (Kim et al., 2023) | 19.7 | 57.8 | 10.2 | 87.2 |
| AdaPlanner (Sun et al., 2023) | 15.1 | 74.1 | 6.1 | 90.3 |
| AdaPlanner (one demo) | 13.1 | 24.4 | 4.5 | 74.3 |
| AutoManual (Chen et al., 2024) | 23.2 | **91.1** | 4.6 | 95.2 |
| ReAct[†] | 16.2 | 84.4 | 9.2 | 92.8 |
| **AutoRPA (code only)** | **1.0** | 80.0 | **0.9** | 92.5 |
| **AutoRPA** | 1.4 | **91.1** | 1.4 | **95.4** |

*Table 3.* Ablation study on key components of AutoRPA on AndroidWorld.

| Variants of AutoRPA | Success (%) |
|---|---|
| **AutoRPA** | **51.7** |
| Building w/o ReAct | 32.5 |
| Building w/o Translator | 40.2 |
| Code Repair w/o ReAct | 45.5 |
| Builder w/o RAG | 48.8 |

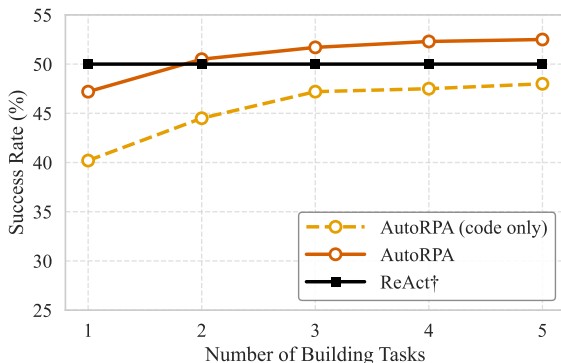

*Figure 5.* The success rate curve of varying building task numbers with GPT-4.1 on AndroidWorld.

ries; furthermore, due to the randomness of LLM outputs, tasks that were previously successful may not necessarily succeed again. In contrast, the RPAs generated by AutoRPA exhibit high stability and robustness to testing tasks. Notably, "AutoRPA (code only)" achieves a comparable success rate to ReAct[†] during testing by executing only the generated RPA code, while consuming minimal tokens (used by grounder in `find_element` and `ask_mllm`). This indicates that AutoRPA can generate efficient and robust RPA scripts for most tasks that ReAct can accomplish.

**Main Results on WebArena.** The real-world web tasks in WebArena are highly challenging, and different tasks within the same task type often require distinct strategies. This necessitates the integration of various strategy logics into a single robust function. As shown in Fig. 4, AutoRPA and "AutoRPA (Code-only)" achieve performance on par with previous methods, while demonstrating a significant reduction in token consumption.

**Main Results on MiniWoB++.** MiniWoB++ has greater randomness in environmental states and task instructions, especially in the 9 types of hard tasks. As shown in Tab. 2, AutoRPA exceeds the previous methods based on ReAct and skill learning, while only consuming less than 10% tokens. Meanwhile, "AutoRPA (code only)" achieves a comparable success rate to ReAct[†]. It is worth noting that both AdaPlanner (Sun et al., 2023) and AutoManual (Chen et al., 2024) require more than one carefully crafted demonstration, indicating that they rely on prior knowledge of the environment to generate reasonable code-style plans

or actions. In contrast, thanks to the ReAct exploration phase, AutoRPA can generate robust RPA code for all task types with only one demonstration on a simple task ("`click-button`").

### 4.2. Ablation Study

More experimental results on token costs during building, comparisons with more advanced agents, more analysis, and case studies are provided in the Appendix.

**Key components of AutoRPA.** As shown in Tab. 3, when building without the ReAct agent, i.e., the builder agent directly generates the RPA code, the success rate drops dramatically. The results also show that the translator agent is important for code generation. The result of "Code Repair w/o ReAct" indicates the importance of our hybrid repair strategy. The effect of the RAG mechanism provided to the builder is evidenced by the drop in "Builder w/o RAG".

**The number of building tasks** $N$**.** As shown in Fig. 5, with the increase in the number of building tasks, the test success rates of both AutoRPA and "AutoRPA (code only)" continue to improve. Notably, the performance of "AutoRPA (code only)" gradually approaches that of ReAct[†], indicating that a larger set of building tasks helps AutoRPA to generate and verify more robust RPA.

# 5. Conclusion

We proposed AutoRPA, a robust framework for automating RPA script synthesis from GUI agent interactions. By introducing translator and builder agents, our system can transform the hard-coded ReAct trajectory into adaptable RPA scripts. Additionally, our hierarchical RAG and hybrid repair strategy enable reliable code refinement. Experimental results across three GUI environments show that AutoRPA achieves competitive success rates compared to SOTA agents, while significantly reducing token usage.

# Impact Statement

This paper presents work whose goal is to advance the field of machine learning. There are many potential societal consequences of our work, none of which we feel must be specifically highlighted here.

# Acknowledgement

This work was supported in part by the National Natural Science Foundation of China under Grants (No. 62502135, 62422204), the Zhejiang Provincial Natural Science Foundation of China under Grants (No. LQN25F030014, LRG26F020001), the Key Research and Development Program of Zhejiang Province (No. 2025C01026), the Scientific Research Innovation Capability Support Project for Young Faculty.

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

## A. Limitations

Although the AutoRPA framework has made significant contributions, there are still several limitations worth further discussion. Firstly, our approach largely relies on the capabilities of advanced MLLMs such as GPT-4.1 to generate ReAct trajectories and RPA code, which may limit the applicability of this framework to less advanced models.

Secondly, our method has certain requirements for the GUI environment: 1. In addition to screenshots, the environment needs to provide the DOM or accessibility tree, so that we can dynamically locate elements based on their attributes. However, this limitation may be alleviated by current advanced screenshot parsing technologies, such as OmniParser-V2 (Lu et al., 2024). 2. During the RPA building stage, users need to provide more than one task that belongs to one task type and the final reward $r$ to obtain an ideal RPA. We plan to address this in future work by having LLM agents generate rewards and autonomously explore tasks, similar to what Voyager (Wang et al., 2024a) has done.

Thirdly, the ReAct agent exhibits limited capability for complex and challenging tasks during the exploration stage. It may be combined with more sophisticated methods, such as tree search algorithms (Putta et al., 2024), to enhance the exploration capability of the ReAct agent for difficult tasks.

Finally, the Builder agent still shows a significant gap compared to humans in terms of tool usage and code generation. For example, it sometimes fails to use the `fetch_info` tool when it should do so to generate code more cautiously; and at other times, it overuses the `fetch_info` tool when it's not necessary. Additionally, it opts for unnecessarily complicated solutions in certain situations where it should use the `ask_mllm` action to obtain information directly. This will be an obstacle to generating RPAs for more complex tasks.

## B. More Implementation Details

Following the standard test setting on AndroidWorld (Rawles et al., 2025), we test on *task0* for each task type. During the testing on Miniwob++, we randomly sample 5 distinct tasks (different from $N$ building tasks). For WebArena (Zhou et al., 2024), we select task types with 4 instances to enable a building phase (3 instances) and testing phase (1 instance), resulting in 19 task types. For all benchmarks, we only provide the agents with a simple demonstration to illustrate the output format.

The maximum number of steps for a ReAct agent to complete a task in AndroidWorld is 10 times the task complexity (capped at 50), while in MiniWoB++ it is 20 steps, and in WebArena 25 steps. For building and testing tasks of a certain task type, we can automatically generate different instances in the environment by controlling the random number seed. For example, if there are 3 building tasks, we use the tasks generated with seed=$1, 2, 3$ for building. In AndroidWorld, the task generated with seed=$0$ is used for testing, while in MiniWoB++, the tasks generated with seed=$0, 11, 12, 13, 14$ are used for testing.

During the testing stage, when evaluating "AutoRPA", we first check whether there exists a fully verified RPA function for the given task type; if not, we then check whether the task type can be accomplished using ReAct. When evaluating "AutoRPA (code only)", we check whether there exists a fully verified RPA function for the given task type; if not, we choose the RPA function that has passed the most verification tasks during the RPA building process.

For SeeAct and M3A, we use their implementation by AndroidWorld (Rawles et al., 2025). For RCI, AdaPlanner, and AutoManual, we use their official implementations on MiniWoB++. As described in Section 3.4, under the default settings of AutoRPA, for tasks that cannot be verified by refining RPA code, the ReAct agent will be executed during testing. To demonstrate the effectiveness of the generated RPA, **AutoRPA (code only)** will only execute the RPA code that has been successfully validated during testing.

In MiniWoB++, for RCI (Kim et al., 2023), AdaPlanner (Sun et al., 2023), and AutoManual (Chen et al., 2024), the agents are provided with 104, 38, and 4 expert demonstrations, respectively, according to their official code on GitHub.

## C. More Experiments and Results

### C.1. Full Experimental Results

**Testing Success Rate of Each Task Type.** The task success rate of each task type in MiniWoB++ is presented in Fig. 6. AutoRPA outperforms ReAct[†] on almost all task types, except for the `multi-layouts` category, where the code

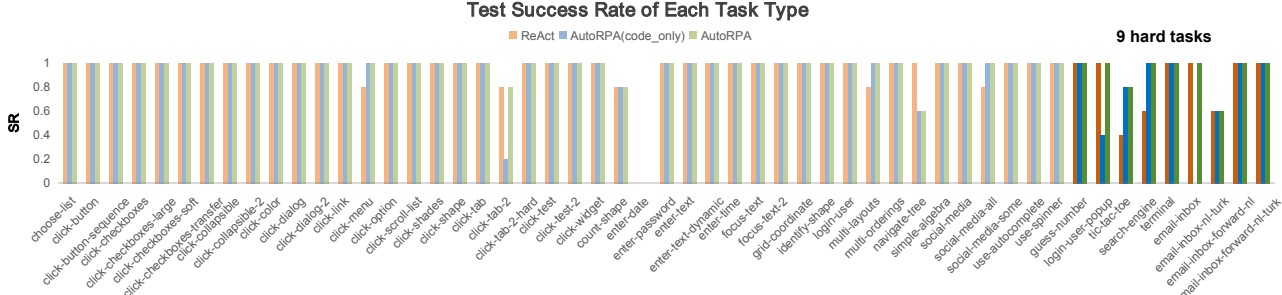

*Figure 6.* The task type-level performance of ReAct[†], AutoRPA (code only), and AutoRPA with GPT-4.1 in MiniWoB++

*Table 4.* Token usage (in thousands) per task type of each module during the building process in AndroidWorld and MiniWoB++, using GPT-4.1.

| Module | AndroidWorld | MiniWoB++ | |
|---|---|---|---|
| | | Hard (9 types) | All (53 types) |
| ReAct (w. reflection) | 164 | 38 | 20 |
| Translator | 37 | 12 | 7 |
| Builder | 20 | 12 | 9 |
| RPA Verification | 12 | 7 | 6 |
| Total | 233 | 69 | 42 |

generated by AutoRPA successfully passes the verification of the building tasks but fails to generalize to all testing tasks.

**Token Cost for Building RPA.** In Tab. 4, we list the tokens consumed by AutoRPA during the building stage. We find that the main source of consumption comes from the exploration process of ReAct, especially in high-difficulty tasks. It is important to note that the overhead for the building process is a one-time cost per task type; once the RPA code is built, it can be used to reduce expenses. Furthermore, 'online building' is feasible for new tasks: the RPA code is generated immediately after the task is successfully completed using ReAct. The initial building overhead is typically amortized after the deployed RPA code executes approximately four tasks.

**Comparison with More Advanced Methods.** In Tab. 5, we compare our results against state-of-the-art GUI Agents, demonstrating that our method remains competitive. It is crucial to note that **AutoRPA is ReAct-agnostic**. As detailed in the main text, our framework does not rely on any specific ReAct agent. In fact, many powerful agents, such as LLMs fine-tuned for 'Computer Use' scenarios (DeepMind, 2025; Wang et al., 2025a), are compatible with the ReAct paradigm. To further demonstrate this, we experimented by replacing the LLM backbone of both ReAct and AutoRPA with a more powerful reasoning model. As shown in the table, utilizing the powerful Claude-sonnet-4.5 boosts the success rates for both methods, while AutoRPA consistently retains its advantage in token efficiency. Furthermore, we experimented with the ReAct Agent that operates solely on screenshots, outputting actions via screen coordinates. The Translator in our method successfully converts these hard-coded actions into UI element decision logic, ultimately generating high-quality RPA code. This further underscores that our AutoRPA framework is orthogonal to the choice of ReAct Agent.

**Statistical Results of Generated RPA.** In Tab. 6, we present the number of verified RPAs generated by AutoRPA for each benchmark. We find that the generated RPA has a high success rate on the testing tasks, indicating its robustness to task instructions. Moreover, generating RPA typically requires about one round of refinement.

## D. More Ablation Studies

**RPA with Affordable GUI Grounder** The `find_element` function relies on a GUI grounding model to locate elements based on descriptions when element attributes alone are insufficient to uniquely identify a GUI element. To further reduce the costs during testing, we also tried replacing the GUI grounding model used in `find_element` and `ask_mllm` during testing with a more affordable MLLM model. In Tab. 8, we replace the default grounding model, i.e., GPT-4.1, with GPT-4o-mini or Qwen2.5-VL-72B-Instruct when evaluating the generated RPA. We find that using a weaker GUI

*Table 5.* Testing success rate and token consumption of LLM agent methods on AndroidWorld.

| Method | Model | Observation Type | Tokens ($k$) $\downarrow$ | Success (%) $\uparrow$ |
|---|---|---|---|---|
| SeeAct (Zheng et al., 2024) | GPT-5 | Screenshot+A11y tree | 134.5 | 40.3 |
| M3A (Rawles et al., 2025) | GPT-5 | Screenshot+A11y tree | 164.6 | 57.0 |
| Agent S3 (Gonzalez-Pumariega et al., 2025) | GPT-5 | Screenshot | - | 68.1 |
| Gemini-2.5-CU (DeepMind, 2025) | Gemini 2.5 Computer Use | Screenshot | - | 69.7 |
| Mobile-Agent-v3 (Ye et al., 2025) | GUI-Owl-32B | Screenshot | - | 73.3 |
| ReAct[†] | GPT-5 | Screenshot+A11y tree | 142.5 | 74.1 |
| ReAct[†] | Claude-sonnet-4.5 | Screenshot | 145.1 | 69.0 |
| ReAct[†] | Claude-sonnet-4.5 | Screenshot+A11y tree | 146.0 | 76.7 |
| **AutoRPA (code only)** | GPT-5 | Screenshot+A11y tree | 6.2 | 70.7 |
| **AutoRPA** | GPT-5 | Screenshot+A11y tree | 30.6 | 75.9 |
| **AutoRPA** | Claude-sonnet-4.5 | Screenshot | 33.5 | 70.1 |
| **AutoRPA** | Claude-sonnet-4.5 | Screenshot+A11y tree | 36.1 | 76.9 |

grounding model results in a decrease in success rate.

**The maximum rounds of code refinements $M$ and reflection $N_{\text{ref}}$.** As shown in Tab. 7, with more refinement turns, the builder can have more chances to generate desirable RPA to pass all verification. With more reflection rounds during the building, the ReAct agent can produce more successful trajectories, thereby strengthening the subsequent RPA generation.

**Unify ReAct and Translator Agents** In AutoRPA, after obtaining a hard-coded ReAct trajectory from the ReAct agent, a Translator agent is employed to convert each action into a soft-coded format that aligns with the current UI elements on the page. We refer to this setup as the **Separate Mode**. We also explore an alternative configuration, where the ReAct and Translator functionalities are unified. In this **Unified Mode**, the ReAct Agent directly outputs both the hard-coded action for immediate execution and the corresponding soft-coded action, thereby eliminating the need for a separate translator agent. Our experiments (see Tab. 9) reveal that generating soft-coded actions directly from the planner increases the input context length, which can slightly degrade the quality of the generated actions.

*Table 6.* The number of generated RPA, testing Success Rate (SR) of generating RPA, and average refinement turn in AndroidWorld and MiniWoB++ using GPT-4.1.

| Task Type | RPA Num (%) | SR of RPA (%) | Refinement Turn |
|---|---|---|---|
| ***AndroidWorld*** | | | |
| All | 53.4 | 88.3 | 1.17 |
| ***MiniWoB++*** | | | |
| Hard (9 types) | 77.8 | 91.4 | 0.89 |
| All (53 types) | 92.5 | 97.6 | 0.85 |

*Table 7.* Ablation study on the number of code refinement turns $M$ and reflection rounds $N_{\text{ref}}$ during the building stage of AutoRPA. We report the success rate (%) on 9 hard and all 53 task types in MiniWoB++ with GPT-4.1. * denotes the default setting.

| Refinement Turns $M$ | AutoRPA (code only) | | AutoRPA | |
|---|---|---|---|---|
| | Hard (9 types) | All (53 types) | Hard (9 types) | All (53 types) |
| 0 | 64.4 | 73.6 | 76.2 | 86.8 |
| 1 | 68.8 | 83.7 | 80.0 | 89.8 |
| 2 | 78.7 | 91.5 | **91.1** | **95.4** |
| 3* | **80.0** | **92.5** | **91.1** | **95.4** |
| **Reflection Rounds $N_{\text{ref}}$** | | | | |
| 0 | 71.8 | 81.5 | 84.4 | 91.7 |
| 1 | 75.0 | 85.3 | 87.4 | 93.2 |
| 2* | **80.0** | **92.5** | **91.1** | **95.4** |

*Table 8.* Performance comparison of different GUI Grounders on RPA Quantity and Success Rate (SR) in AndroidWorld and MiniWoB++. Task seed = 0.

| Grounder | SR of RPA (%) ↑ |
|---|---|
| ***AndroidWorld*** | |
| GPT-4o-mini | 79 |
| Qwen2.5-VL-72B-Instruct | 83.8 |
| GPT-4.1 | 88.3 |
| ***MiniWoB++*** | |
| GPT-4o-mini | 95.9 |
| Qwen2.5-VL-72B-Instruct | 97.6 |
| GPT-4.1 | 97.6 |

*Table 9.* Comparison of Separate and Unified Planner-Translator Architectures in AndroidWorld and MiniWoB++. * denotes the default setting in AutoRPA.

| Mode | RPA Num (%) ↑ | SR of RPA (%) ↑ | AutoRPA(code) (%) ↑ |
|---|---|---|---|
| ***AndroidWorld*** | | | |
| Unified | 53.4 | 80.6 | 43.1 |
| Separate* | 53.4 | 88.3 | 47.2 |
| ***MiniWoB++*** | | | |
| Unified | 96.2 | 88.6 | 85.3 |
| Separate* | 92.5 | 97.6 | 92.5 |

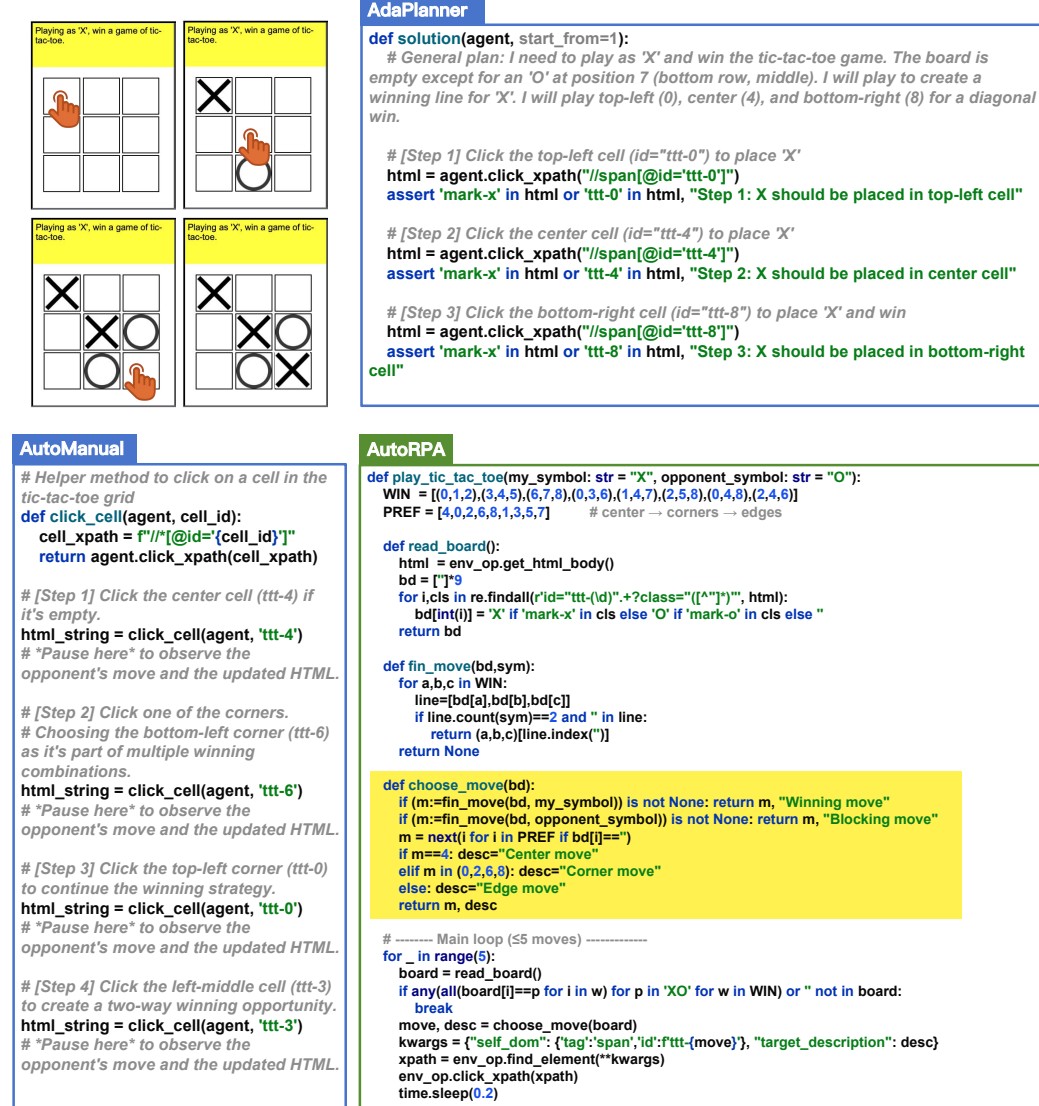

*Figure 7.* Code generated by different agent frameworks for *tic-tac-toe*. To facilitate a clear comparison of the core differences, we present a streamlined version of the code, generated by AutoRPA, that preserves all essential decision logic.

# E. Case study of AutoRPA

### E.1. Generated Code: AutoRPA vs. Prior Methods

We use the *tic-tac-toe* task type in MiniWoB++ as a case study to compare the skill code produced by prior methods, i.e., AdaPlanner (Sun et al., 2023) and AutoManual (Chen et al., 2024), and the RPA code generated by AutoRPA.

The task objective is: `Playing as 'X', win a game of tic-tac-toe.` The opponent is a computer program, and the actions of that computer program have a certain degree of randomness. Therefore, using the exact same steps does not always guarantee success.

**AdaPlanner (Sun et al., 2023).** See Fig. 7. The code adopts a *static three-step plan*: it always clicks the top-left cell (`ttt-0`), the centre (`ttt-4`), and the bottom-right (`ttt-8`) in sequence, aiming to form the main diagonal `XXX`. After each action, the code immediately issues an `assert` to verify that the expected number of `X` marks is present.

This design showcases AdaPlanner's *Plan-then-Execute with Skill Learning* paradigm: the code is extremely concise (no loops, no board parsing) and provided the opponent does not block the diagonal win in the minimum three moves. The fixed skill code produced by AdaPlanner is likely obtained when a particular attempt happens to succeed. However, the

policy is brittle: it neither reacts to any opponent move that interferes with the pre-planned line. So the first unexpected state will trigger an assertion failure and abort the run. Then Adaplanner will need to call LLM to fix it in a closed-loop manner. Consequently, while highly efficient in favorable conditions, the code lacks the adaptability of the dynamic parsers and defensive heuristics.

**AutoManual (Chen et al., 2024).** See Fig. 7. AutoManual resolves the Path Dependency issues found in Adaplanner. It takes into account that the generated skill code serves as a prompt for the LLM, and thus, after a successful attempt by ReAct (with code-style actions), it adds "*Pause here*" to remind future attempts. However, it is unable to escape the constraints of *Skill Learning for ReAct* paradigm, resulting in that the generated skill code cannot be executed directly and relies on the LLM as the decision maker.

**AutoRPA (ours).** See Fig. 7. Unlike the prior methods, the code generated by our AutoRPA is more complicated and can automatically cope with various scenarios. The function repeatedly parses the DOM via regular expressions that recognise both `class` attributes (`mark-X`/`mark-O`) and literal text content, offering strong robustness to UI variations.

Within the loop for player `X`, the script follows a clear priority queue:

(1) Win: take any move that completes three in a row.
(2) Block: intercept any opponent's winning move.
(3) Centre: occupy the middle cell if free.
(4) Corner: choose an empty corner.
(5) Edge: fall back to a side cell.

Each action is executed through `env_op.find_element` followed by `env_op.click_xpath()`, and then waits briefly for the UI to refresh before re-evaluating the board.

Compared with other code, this RPA offers (1) greater HTML robustness via multi-pattern parsing and retries, (2) a complete self-contained game loop capable of finishing an entire match, and (3) parameterised symbols and abstracted DOM access, making it easier to port to other MiniWoB++ tasks or different player orders.

### E.2. Examples of RPA Building Process

We present an example of the RPA building process using the MarkorCreateNote task from the AndroidWorld benchmark. The task instances of the task type are generated from the template: `Create a new note in Markor named {file_name} with the following text: {text}`.

**Exploration Phase:** As illustrated in Fig. 8, through the ReAct agent and the translator agent, we can obtain a hard-coded trajectory and a soft-coded trajectory.

**RPA Generation and Refinement Phase:** After completing a building task with the ReAct agent, we employ a builder agent to generate a robust RPA function for the task type. See RPA Function Info in Code Block - Initial RPA Code on MarkorCreateNote. In this example, the program was interrupted during the testing of the generated RPA function (See Fig. 9). According to the analyzer's diagnosis, the function failed to extract the file name during execution. Furthermore, since the file suffix was changed from the default .md to .txt, it also became necessary to explicitly set the file suffix. Based on the analyzers recommendation, the ReAct agent resumes execution from the current screen and performs actions such as entering the file name and specifying the file suffix. After that, the Builder will generate the refined code based on the hybrid trajectory. See refined code in Code Block - Refined RPA Code on MarkorCreateNote.

**Task 1:** Create a new note in Markor named *final_quick_eagle.md* with the following text: *'Winter is coming.'*

*Figure 8.* ReAct Trajectory in AndroidWorld – MarkorCreateNote

**Task 2:** Create a new note in Markor named *quick_umbrella_2023_04_27.txt* with the following text: *'Dry cleaning is ready for pick-up.'*

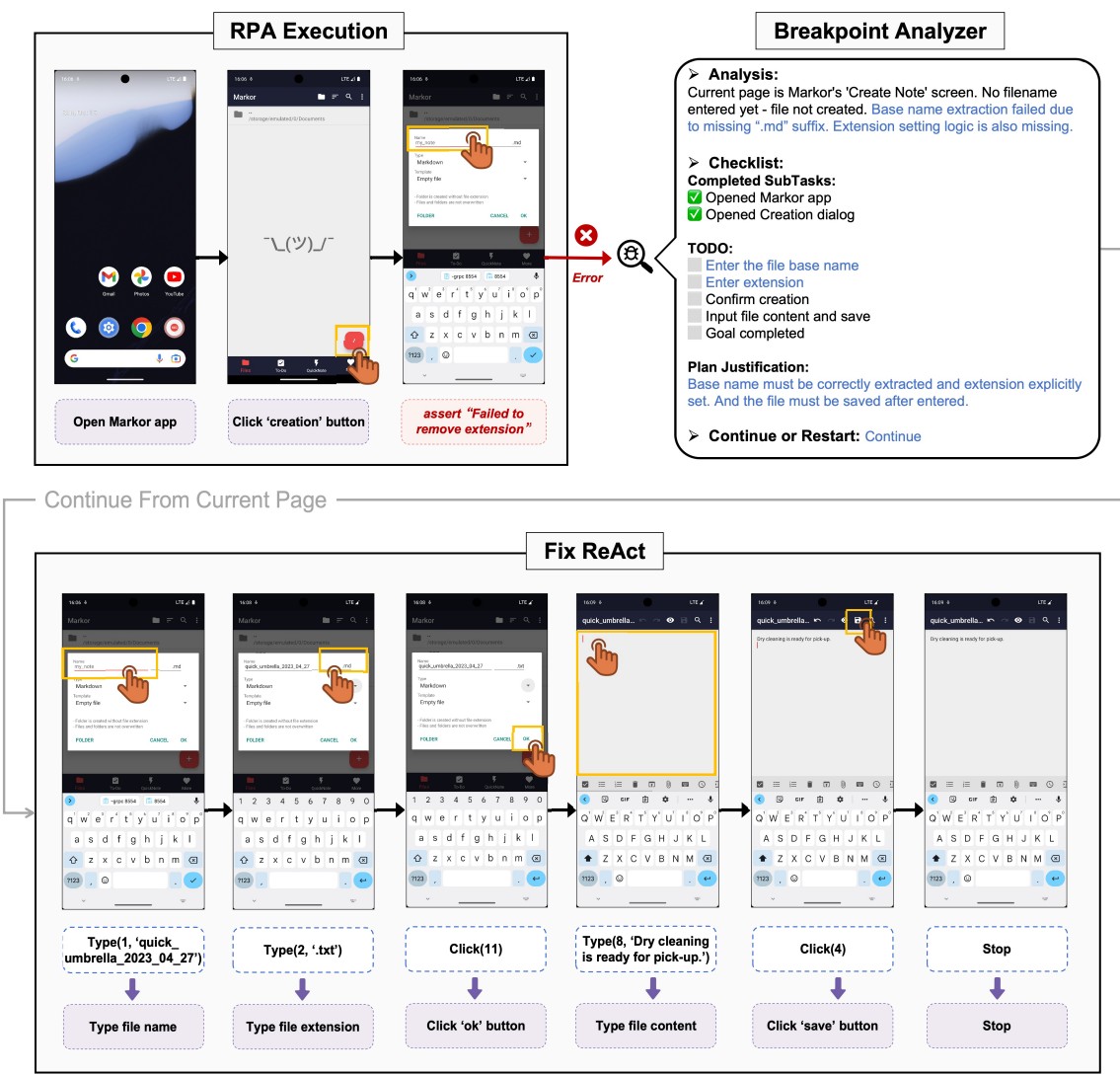

*Figure 9.* Hybrid Repair Process in AndroidWorld – MarkorCreateNote

```python
code :  Initial RPA Code on MarkorCreateNote

### Func Description:
# Create a new note in Markor with a specified name and content.

### Params Description:
# - file_name (Optional[str]): The name of the note to create (with or without extension).
# - text (Optional[str]): The content to write into the note.

### Example Usage:
# create_markor_note(file_name="final_quick_eagle.md", text="Winter is coming.")

### RPA Code:
def create_markor_note(file_name: str = None, text: str = None):
    # Step 1: Open the Markor app
    env_op.open_app('Markor')

    # Step 2: Click the "+" button to create a new file/folder
    retry = 0
    max_retry = 3
    plus_index = -1
```

```python
    while retry < max_retry:
        kwargs = {
            "content_description": "Create a new file or folder",
            "additional_actions": ["long_press"],
            "target_description": "Red circular '+' button at the bottom right to create a new file or folder"
        }
        plus_index = env_op.find_element(**kwargs)
        if plus_index != -1:
            env_op.click(plus_index)
            break
        else:
            # Try swiping up in case the button is off-screen
            env_op.swipe("up")
            retry += 1
    assert plus_index != -1, "Failed to find '+' button to create new file."

    # Step 3: Input the file name in the dialog
    retry = 0
    max_retry = 3
    while retry < max_retry:
    kwargs = {
        "hint_text": "my_note",
        "additional_actions": ["input_text"],
        "target_description": "Input field for file name in the dialog"
    }
    name_index = env_op.find_element(**kwargs)
    if name_index != -1:
        assert file_name is not None and file_name.endswith('.md'), "Failed to remove extension"
        base_name = file_name[:-3]
        env_op.input_text(base_name, name_index, True)
    assert name_index != -1, "Failed to find file name input field."

    # Step 4: Click "OK" to confirm creation
    retry = 0
    max_retry = 3
    ok_index = -1
    while retry < max_retry:
        kwargs = {
            "text": "OK",
            "target_description": "Confirmation button to create the new file in the file/folder creation
            ↪   dialog"
        }
        ok_index = env_op.find_element(**kwargs)
        if ok_index != -1:
            env_op.click(ok_index)
            break
        else:
            env_op.wait()
            retry += 1
    assert ok_index != -1, "Failed to find OK button to confirm file creation."

    # Step 5: Input the note content
    retry = 0
    max_retry = 3
    content_index = -1
    while retry < max_retry:
        kwargs = {
            "additional_actions": ["long_press", "input_text"],
            "target_description": "Large central text input area for editing the note content"
        }
        content_index = env_op.find_element(**kwargs)
        if content_index != -1:
            env_op.input_text(text or "", content_index, True)
            break
        else:
            env_op.wait()
            retry += 1
    assert content_index != -1, "Failed to find note content input area."

    # Step 6: Save the file
    kwargs = {
        "content_description": "Save",
        "target_description": "Save button to save the current file in the upper right corner"
    }
    save_index = env_op.find_element(**kwargs)
    if save_index != -1:
        env_op.click(save_index)
    assert save_index != -1, "Failed to find Save button to save file."

    # Step 7: Mark task as complete
```

Python

```python
    env_op.stop("complete")
```
*Python*

---

## code :  Refined RPA Code on MarkorCreateNote

```python
### Func Description:
# Create a new note in Markor with a specified name, file type, and content, ensuring the note is saved.

### Params Description:
# - file_name (Optional[str]): The desired name of the note file (with or without extension).
# - file_extension (Optional[str]): The file type/extension (e.g., "txt", "md").
# - text (Optional[str]): The content to be written into the note.

### Example Usage:
# create_markor_note(
#     file_name="quick_umbrella_2023_04_27.txt",
#     file_extension="txt",
#     text="Dry cleaning is ready for pick-up."
# )

### RPA Code:
def create_markor_note(file_name: str = None, file_extension: str = None, text: str = None):
    # Step 1: Open the Markor app
    env_op.open_app('Markor')

    # Step 2: Click the "+" button to create a new file/folder
    for _ in range(3):
        kwargs = {
            "content_description": "Create a new file or folder",
            "target_description": "red circular button with plus sign at bottom right to create a new file or
            ↪  folder"
        }
        index = env_op.find_element(**kwargs)
        if index != -1:
            env_op.click(index)
            break
        env_op.wait()
    assert index != -1, "Create new file button not found"

    # Step 3: Input the file name (without extension)
    for _ in range(3):
        kwargs = {
            "text": "my_note",
            "hint_text": "my_note",
            "additional_actions": ["input_text"],
            "target_description": "Input field for the file name in the new file/folder creation dialog"
        }
        name_index = env_op.find_element(**kwargs)
        if name_index != -1:
            # Remove extension if present in file_name
            base_name = file_name
            if file_name and '.' in file_name:
                base_name = file_name.rsplit('.', 1)[0]
            env_op.input_text(base_name or "untitled", name_index, True)
            break
        env_op.wait()
    assert name_index != -1, "File name input field not found"

    # Step 4: Set the file type (extension) if specified
    kwargs = {
        "text": ".md",  # Default extension, may vary
        "hint_text": ".md",
        "additional_actions": ["input_text"],
        "target_description": "Input field for extension, right of the file name"
    }
    ext_index = env_op.find_element(**kwargs)
    if ext_index != -1:
        if not file_extension.startswith('.'):
            file_extension = '.' + file_extension
        env_op.input_text(file_extension, ext_index, True)
    assert ext_index != -1, "File extension input field not found"

    # Step 5: Confirm creation ("OK" button)
    for _ in range(3):
        kwargs = {
            "text": "OK",
            "target_description": "Confirmation button to create the new file in the file creation dialog"
        }
        ok_index = env_op.find_element(**kwargs)
        if ok_index != -1:
```
*Python*

```python
                env_op.click(ok_index)
                break
        env_op.wait()
    assert ok_index != -1, "OK button not found"

    # Step 6: Input the note content
    for _ in range(3):
        kwargs = {
            "additional_actions": ["long_press", "input_text"],
            "target_description": "Large empty text input area in the center of the Markor note editor for
↪   editing note content"
        }
        editor_index = env_op.find_element(**kwargs)
        if editor_index != -1:
            env_op.input_text(text or "", editor_index, True)
            break
        env_op.wait()
    assert editor_index != -1, "Note editor input area not found"

    # Step 7: Ensure the note is saved before exiting
    for _ in range(3):
        kwargs = {
            "content_description": "Save",
            "target_description": "Save button (floppy disk icon) in the note editor toolbar"
        }
        save_index = env_op.find_element(**kwargs)
        if save_index != -1:
            env_op.click(save_index)
            env_op.wait()
            break
        else:
            # If not found, wait and retry
            env_op.wait()
    # No assertion: Save button may disappear if already saved

    # Step 8: Mark task as complete
    env_op.stop("complete")
```
`Python`

## F. Prompts

### F.1. Prompts for Agents Generating ReAct Trajectory

Our GUI agent framework builds upon the ReAct paradigm, with customized prompt engineering to better support GUI-based tasks. The system consists of three specialized components: (1) a ReAct agent for step-by-step planning and next-action generation, (2) a summarizer agent for reporting the execution outcomes of individual actions, and (3) a concluder agent for summarizing the entire task trajectory. We list their system prompts in Lst. 1, Lst. 2, and Lst. 3 respectively.

*Listing 1.* System Prompts for ReAct Agent

```
[Role]
You are an intelligent agent that operates a GUI for a user.
Your role is to break down complex requests into optimal, step-by-step actions and execute them while adapting to
 changes.

[ADMISSIBLE ACTIONS]
{Env Operations}

[Output Format]
Your output must strictly follow the structure below. Headings must match exactly, including `###`, spacing, and
 colons.

### Observations:
Summarize key input details and immediate screen observations. Include all obvious insights.
### Completed Tasks:
List completed tasks, each starting with a ✓.
### Plan Justification:
Briefly explain the rationale behind your plan.
### Plan List:
List tasks to achieve the goal, each starting with a ❑; if the goal is achieved, output "goal completed."
### Next Action Justification:
Explain why the next action is chosen.
### Action:
Output the next action between '```python' and '```'(one action, one line). Use only [ADMISSIBLE ACTIONS].

[Your Workflow]
1. Analyze Input: Carefully examine all input. Extract all goal-relevant info dont miss anything useful. Directly
```

```
 infer and record any obvious conclusions.
2. Evaluate Progress: Check if the goal is achieved; if so, stop. Otherwise, update completed tasks.
3. Devise Plan: Break the goal into efficient, non-redundant steps. Identify and obtain any missing information.
4. Execute & Adjust: Analyze the UI info to decide actions. Adjust the plan if elements are missing or inaccessible.
5. Error Handling: Retry once on failure; if it still fails, choose an alternative.
6. Generate Next Action: Choose the next logical action that advances the goal.

[Guidelines]
Follow these guidelines:
- After you output the action, the action will be executed. The results of each action and the new observations will
  be printed to you at next step.
- Maintain a holistic view by identifying the specific steps required to complete the task using the current input.
- Fully leverage provided input to reduce unnecessary follow-up actions.
- Interact only with verified UI elements.
- Adapt dynamically adjust to screen changes and execution failures.

Operation Guidelines:
- Choose the simplest method.
- Ensure the index for click, long_press, and input_text is visible in both the screenshot and UI list.
- Swipe to reveal hidden UI elements.
- Confirm that a UI element supports the intended action before interacting.
- Every element is clickable by default.
- Use standard text selection methods (long-press and selection bar) when needed.
```

*Listing 2.* Prompt for Summarizer Agent

```
[Role]
You are an ActionSummarizer agent. Evaluate the success of actions executed on an Android device by comparing '
 before' and 'after' screenshots.

[Output Format]
Your output should consist of the following things:
### Screen Changes:
Summarize the primary differences between the before and after screens (max 30 words). If none, respond with "
 Nothing Happens."
### Execution Summary:
In one concise line (max 50 words), state whether the page changed as expected, including the intent, outcome, and
 key insights.

[Your Workflow]
1. Compare Screenshots: Focus on differences related to the highlighted element in the 'before' screenshot and the
 executed code.
2. Verify Purpose: Check if the executed code aligns with its intended purpose (reason for code) and if the
 highlighted element meets expectations.
3. Compare Code: Confirm that the expected code matches the executed code; if not, identify discrepancies.
4. Assess Outcome: Determine if the executed code met the intended goal.
5. Highlight Findings: Note key insights for future actions.

[Guidelines]
- If actions like `answer` or `wait` do not change the screen, assume success.
- If no change occurs, clearly state the failure and possible reasons.
- Rely primarily on screenshot analysis.
- Focus on actionable insights; avoid redundant details.
- For file-related operations, make sure to operate only on the exact target file; do not interact with similar
  files. You must locate and use the precise file required.
- When naming files, ensure proper file extensions.
```

*Listing 3.* Prompt for Concluder Agent

```
[Role]
You are a concluder agent tasked with summarizing a failed screen operation trajectory and performing a critical
 reflection to prevent future errors.

[Formatting Guidelines]
Your output must strictly follow the structure below. Headings must match exactly, including `###`, spacing, and
 colons.
Required sections (in order):
### Episode Conclusion:
### Reflection:

[Output Format]
### Episode Conclusion:
Provide a summary of the trajectory, clearly indicating where the failure occurred.
### Reflection:
Identify and describe the key actions that failed, including specific reasoning for each failure (e.g., incorrect
 actions, UI misunderstandings, planner's incorrect judgment). State the primary cause(s) of failure. Provide clear,
  actionable steps to address these issues and achieve the task.
```

```
[Your Workflow]
1. Analyze Trajectory:
  - Identify and pinpoint exactly which step in the trajectory led to failure.
  - Clearly explain why this specific step failed (e.g., incorrect actions, misinterpretation of UI, planners
    inaccurate decision-making).
  - Highlight key decision points and provide specific reasoning behind each critical action.
2. Root Cause Analysis (RCA):
  - Clearly state the underlying cause(s) of the failure.
  - Highlight any misjudgments or missed opportunities for correction.
3. Formulate Corrective Guidelines:
  - Propose clear, actionable guidelines or improvements for avoiding similar failures in future attempts.
4. Summary Generation:
  - Focus on key actions that directly contributed to the goal, showing how each step led to the next.
  - Highlight the reasoning behind critical decisions and their role in the task's success.
  - Write a single coherent paragraph in natural language, emphasizing the causal relationships between actions.

[GUIDELINES]
- Avoid generic or unrelated descriptions; strictly avoid irrelevant or overly broad content.
```

## F.2. Prompts for Translator Agent

*Listing 4.* Prompt for Translator Agent

```
[System]
Generate a Soft-coded Action by dynamically replacing the hardcoded index in the given action with an element-
 matching strategy.
Before that, determine from the observation and action justification whether the original action can be decided by
 mechanical element matching; if not, consider using `ask_mllm()` for combined visual and language reasoning.
MLLMs are inherently random, so `ask_mllm` may not always return expected results. To ensure stability, use strict
 prompt constraints or minimize their usage.

If using `find_element()`, ensure that:
1. The revised logic maintains the same intended behavior as the original hardcoded action;
2. If indexing is not required, do not use the find_element method.

[Index Replacement]
You need to use this function to replace the hardcoded `index` value with the index variable generated by the `
 find_element()`.
### Get Element Index
env_op.find_element(**kwargs) -> int # Use this function to find an element in the UI list using filtering criteria
 and return its index. If no matching element is found, the index will be -1.
- Rules for generating `kwargs`:
  - Use find_element only when an index is required (e.g., not for env_op.open_app).
  - Extract all attributes necessary to uniquely identify the element in the current page. Keys must be correct, and
    values must be fully preserved without modification. For additional_actions, include only the necessary action(s
   ), not the full list.
  - From text, hint_text, content_description, and tooltip, extract only onewhichever is most relevant. This does
    not limit the use of other attributes.
  - Always include target_description: briefly describe the elements role, appearance, position on screen, and any
    dynamic/contextual content that helps make it unique.
- `kwargs` Examples:
  - kwargs = {"content_description": "Save Changes", "target_description": "rectangle button to save"}

[ADMISSIBLE ACTIONS]
{Env Operations}

# You may additionally consider the following smart action patterns if relevant:
### Ask MLLM to answer one question
Dont rely too much on ask_mllm.
env_op.ask_mllm(question: str, output_format: str) -> str # Will return the answer in plain text.

[Output Format]
Your output must include two sections:
### Thought:
Provide a brief explanation (under 30 words) for how you constructed the dynamic UI element matching parameters (i.e
 . the `kwargs`).
### Soft-coded Action:
- If not using `find_element()`: write new code in proper way.
- If the original action requires a UI element index:
Replace the hardcoded index with a dynamic lookup using `env_op.find_element`. You must output *exactly* these three
 lines of code:
```python
kwargs = {...} # A dictionary describing the target UI element
index = env_op.find_element(**kwargs) # Use env_op to locate the element dynamically
env_op.xxx(...) # Replace xxx with the correct action using the index
```
- If the original action does NOT require an index: Simply output the soft-coded action without calling env_op.
  find_element.
```

## F.3. Prompts for Builder Agent

*Listing 5.* System Prompts for Builder Agent

```
[Role]
You are an expert AI coding assistant specializing in generating Skill Code for RPA (Robotic Process Automation).
Your goal is to analyze multiple execution trajectories and initial Skill Codes to produce robust, flexible, and
 reusable code that reliably completes similar tasks of the provided Task Template.

[ADMISSIBLE ACTIONS]
{Env Operations}

### Get Element Index
env_op.find_element(**kwargs) -> int # Use this function to find an element in the UI list using filtering criteria
 and return its index. If no matching element is found, the index will be -1.
- Rules for generating `kwargs`:
 - Use find_element only when an index is required (e.g., not for env_op.open_app).
 - Extract all attributes necessary to uniquely identify the element in the current page. Keys must be correct, and
   values must be fully preserved without modification. For additional_actions, include only the necessary action(s
  ), not the full list.
 - From text, hint_text, content_description, and tooltip, extract only onewhichever is most relevant. This does
  not limit the use of other attributes.
 - Always include target_description: briefly describe the elements role, appearance, position on screen, and any
  dynamic/contextual content that helps make it unique.
- `kwargs` Examples:
 - kwargs = {{"content_description": "Save Changes", "target_description": "rectangle button to save"}}

### Get Current UI Elements List
env_op.get_cur_ui_element_list() -> list[dict] # Returns: List[Dict] Each dictionary contains properties of a UI
 element on the current page.
- Example return:
 [{{'index': 0, 'additional_actions': ['swipe']}}, {{'index': 1, 'content_description': 'Home'}}, {{'index': 2, '
  text': 'Phone', 'content_description': 'Phone', 'additional_actions': ['long_press']}}]

### Ask MLLM to answer one question
Dont rely too much on ask_mllm.
env_op.ask_mllm(question: str, output_format: str) -> str # Will return the answer in plain text.

[Output Format]
Your output must follow this structure exactly, including headings, spacing, and colons. Required sections (in order
 ):
### Thought: State the failure cause, highlight differences from success, and suggest robustness improvements. Avoid
  vague details. (under 100 words)
### Parameters: Define function parameters from the Task Template. Names must be generic and reusable across tasks.
 Make all parameters Optional to improve generalization and flexibility.
### Skill Description: Summarize the current skill based on the Task Template. (under 30 words)
### Skill Code: Provide a single Python code block (between ```python and ```) with well-commented code ready for
 execution on GUI device.
### Example Usage: Provide a separate Python code block for practical usage. Do not include it in the "### Skill
 Code:" section; place it under "### Example Usage:"
### Conclusion:
Summarize how the Skill Code was constructed, considering screen state, input parameters, and task context (under
 100 words). Explain:
 - How the code adapts to screen and UI changes.
 - How the Task Template or previous Skill Code influenced it.
 - What robustness or generalization strategies were applied.
 - Why its reusable across similar tasks.

## Tool Usage
You can use the following tool to fetch ui_list and screenshot about a specific step in a trajectory.
Only use this tool when it is truly necessary to examine a **critical step** that is blocking your reasoning. Avoid
 unnecessary calls.
**Tool**: `fetch_info(source: str, step: int, info: List[str])`
- `source`: one of `pre_skill_exec_traj`, `successful_react_traj`, `failed_react_traj`, `fix_react_traj`
- `step`: the 1-based index of the step to inspect (e.g., 1, 2, 3...)
When you need extra information, respond **only with a JSON object** in the following format (omit the [Output
 Format]):
```json
{{
  "action": "fetch_info",
  "arguments": {{
    "source": "pre_skill_exec_traj",
    "step": 3
  }}
}}
```

[GUIDELINES]
Follow these guidelines:
- Choose the simplest method.
```

```
- Ensure the index for click, long_press, and input_text is visible in both the screenshot and UI list.
- Swipe to reveal hidden UI elements.
- Confirm that a UI element supports the intended action before interacting.
- Every element is clickable by default.
- Use standard text selection methods (long-press and selection bar) when needed.

Code Writing:
- No need to import env_op.
- Add concise comments explaining key parts of the code.
- When searching or using loops, always include: (1) a retry limit, (2) fallback actions like env_op.go_back() or
  alternative search logic, and (3) assertions to ensure the element is found.
- Do not hardcode task content or page information in the code.
- MLLMs are inherently random, so `ask_mllm` may not always return expected results. To ensure stability, use strict
  prompt constraints or minimize their usage.

Output:
- Avoid generic, irrelevant, or overly broad descriptions.

[Your Workflow]
1. Analyze Trajectories:
  - Review the execution history for beneficial vs. irrelevant steps, even if the task succeeded, to improve
    efficiency.
  - Perform a Root Cause Analysis (RCA) on failed trajectories to identify the exact reasons for failure.
  - Compare successful and failed trajectories, highlighting the differences or weaknesses that need improvement.
  - Propose improvements to the code logic and error-handling, providing a detailed optimization strategy in the
    High-Level Plan.
2. Generate Optimized Skill Code:
  - Wrap the code in a reusable function (e.g., def function_name():) with generic parameters. Ensure the code
    handles all cases.
  - Structure the code based on the High-Level Plan.
  - Implement clear error handling with assertions to identify issues, avoiding internal error catching. Error
    handling is external.
  - Do not alter kwargs values, except for dynamically replacing target_description if needed. Ensure kwargs (if
    used) always includes target_description.
3. Enhance Generalization:
  - Improve logging, readability, and maintainability.
  - Ensure the code is general, reusable, and applicable to similar tasks.
```

## F.4. Prompts for Executor and Analyzer Agents

*Listing 6.* System Prompts for RPA Executor Agent

```
[Role]
You are an expert in extracting task parameters for Android RPA functions.
Your task is to accurately extract the required parameters for a new task, based on the provided skill_description
 and skill_parameters format.

[Input]
skill_description: {skill_description}
skill_parameters: {skill_params}
Example Usage: {skill_example}
New Task: {task}

[Output Format]
Extract the appropriate parameters from the **New Task** according to the skill_parameters specification, and
 construct a function call following the structure in the Example Usage.
**Do not change the function name or any of the parameter names under any circumstances.**

Output only the function call in the following format:
```python
function_name(param1=value1, param2=value2, ...)
```
```

*Listing 7.* System Prompts for Analyzer Agent

```
[Role]
You are an intelligent agent that operates an Android phone for a user.
Examine the current screen and review historical execution data to assess task status.
Evaluate progress, update completed tasks, and devise an efficient plan.
Can the current page continue execution to complete the task? Respond with Y or N.

[ADMISSIBLE ACTIONS]
{Env Operations}

[GUIDELINES]
Follow these guidelines:
- Choose the simplest method.
```

```
- Ensure the index for click, long_press, and input_text is visible in both the screenshot and UI list.
- Swipe to reveal hidden UI elements.
- Confirm that a UI element supports the intended action before interacting.
- Every element is clickable by default.
- Use standard text selection methods (long-press and selection bar) when needed.

[Output Format]
### Observations:
Summarize key input details and immediate screen observations. Include all obvious insights.
### Completed Tasks:
List completed tasks, each starting with a ✓.
### Plan Justification:
Briefly explain the rationale behind your plan.
### Plan List:
List tasks to achieve the goal, each starting with a ❏; if the goal is achieved, output "goal completed."
### Whether To Continue:
Just output one letter -- Y or N.

[Input Provided]
- Task Execution Trajectory:
{exec_trajs_str}

- The current screenshot and its ui elements list:
{ui_list}
```

