# OpenReview forum: "AutoRPA: Efficient GUI Automation through LLM-Driven Code Synthesis from Interactions"
_ICML.cc/2026/Conference — ICML 2026 regular_

### Official Review · Reviewer_ZCz8 · 2026-03-03

**Soundness:** 3
**Presentation:** 3
**Significance:** 3
**Originality:** 3
**Overall Recommendation:** 4
**Confidence:** 4

**Summary:**

This paper proposes AutoRPA, which transforms ReAct-style interaction trajectories into reusable RPA functions for efficient GUI automation. The framework adopts a translator–builder–verifier pipeline, where interaction trajectories are converted into soft-coded processes, synthesized into RPA functions via retrieval-augmented generation, and refined through hybrid repair combining RPA execution with ReAct fallback.

Experiments on AndroidWorld, WebArena, and MiniWoB++ demonstrate substantial token reduction (82%–96%) while maintaining competitive task success rates compared to ReAct-based methods.

**Compliance With Llm Reviewing Policy:**

Affirmed.

**Final Justification:**

Taking into account the opinions of the other reviewers and rebuttals of authors, I decide to maintain my score.

**Key Questions For Authors:**

1. Could the authors provide additional details on how AutoRPA scales as the number of reusable functions increases (e.g., retrieval efficiency and maintenance cost)?

**Limitations:**

Yes. The authors discuss the limitations of the framework and no significant negative societal impact is apparent.

**Strengths And Weaknesses:**

1. **Strengths**

(1) Clear and practical motivation.
The paper identifies the inefficiency of ReAct-style interaction for recurring GUI tasks and proposes transforming interaction trajectories into reusable RPA functions, addressing the trade-off between reasoning cost and deterministic execution.

(2) Well-structured framework design.
The translator–builder–verifier pipeline is clearly organized, including trajectory abstraction, retrieval-augmented code synthesis, and hybrid repair with ReAct fallback, forming a coherent system.

(3) Substantial efficiency gains.
Experiments report significant token reduction (82%–96%) while maintaining competitive task success rates, demonstrating improved efficiency and reusability.

(4) Cross-benchmark validation and ablations. Results on AndroidWorld, WebArena, and MiniWoB++ together with component ablations provide empirical support for the effectiveness of each module.




2. **Weaknesses**

(1) Limited richness of the main experimental comparisons.
The primary results emphasize token reduction and task success rate, but the evaluation could be strengthened with more comprehensive metrics (e.g., latency, wall-clock execution time, scalability across tasks) and broader comparisons with additional recent GUI agent baselines.

(2) Dependence on strong backbones and structured environments.
The framework relies on powerful multimodal LLMs and structured signals such as DOM or accessibility trees, which may limit generalizability to less structured or noisier GUI settings.

(3) Limited theoretical or analytical insight.
While the system design is well motivated, the paper provides limited formal analysis of when RPA synthesis succeeds or fails, or how performance scales with task complexity and trajectory variability.

---

> ### Author Rebuttal · Authors · 2026-03-30
>
> **(1) Limited richness of experimental comparisons**
>
> We would like to clarify that runtime efficiency is already part of our main evaluation. In Table 1, we report average execution time, token consumption, and success rate on AndroidWorld, not only tokens and SR. More specifically, AutoRPA consistently reduces both execution time and token usage relative to ReAct-style agents while maintaining or improving success rate. In addition, broader comparisons with more recent GUI agents are included in Appendix Table 5, where we compare against stronger and more recent baselines such as Agent S3, Gemini-2.5-CU, and Mobile-Agent-v3. We agree that these results should have been highlighted more clearly in the main paper, and we will move this discussion forward in the revision.
>
> **(2) Dependence on strong backbones and structured environments**
>
> We agree this is an important limitation, and we already discussed it explicitly in Appendix A. At the same time, our results suggest that the method is not tied to one specific backbone: the advantage of AutoRPA persists across GPT-4o, GPT-4.1, GPT-5, and Claude-Sonnet-4.5. Thus, while stronger MLLMs improve both ReAct and AutoRPA, the benefit of distilling reasoning into reusable RPA code is robust across multiple backbones.
>
> For structured signals, we agree that DOM/a11y trees improve stability and efficiency. We already discussed this limitation in Appendix A and noted that screenshot parsing tools such as OmniParser-V2 can alleviate it. To directly address the reviewer’s concern, we extend experiments in Table 5 and additionally evaluated a fully visual version of AutoRPA without any DOM/a11y trees on AndroidWorld using Claude-sonnet-4.5. In this setting, we use a screenshot-only ReAct Agent; the `find_element` and `ask_mllm` functions in both Translator and Builder rely directly on the grounder model for visual parsing and localization. To obtain the screenshot with SoM, we use the UI detector in OmniParser-V2.
>
> | Method | Setting | Test Tokens (k) | Test SR (%) |
> | :--- | :--- | :--- | :--- |
> | ReAct| Screenshot+A11y tree | 146.0 | 76.7 |
> | ReAct| Screenshot Only | 145.1 | 69.0 |
> | ReAct| Screenshot Only (seed-2.0-pro) | 105.0 | 63.3 |
> | AutoRPA | Screenshot+A11y tree for All Agent | 36.1 | 76.9 |
> | AutoRPA | Screenshot Only for All Agent | 65.6 | 69.8 |
> | AutoRPA | Screenshot Only for All Agent, seed-2.0-pro as grounder | 60.5 | 69.3 |
>
> These results show that removing DOM/a11y indeed increases cost and slightly reduces performance, but AutoRPA remains effective and still uses substantially fewer test-time tokens than ReAct. Furthermore, we can introduce a much cheaper grounder model (e.g., seed-2.0-pro, priced at 1/13 of Claude-sonnet-4.5) during testing while maintaining a comparable success rate. This further supports our claim: ReAct spends most of its test-time budget on repeated "reasoning" rather than "grounding", and AutoRPA distills this reasoning logic into RPA code. We will include these experiments and discussions in the final version.
>
> **(3) Limited theoretical or analytical insight**
>
> We agree that formal analysis would be valuable. Our current paper is primarily an empirical systems paper, and our analytical evidence is therefore empirical rather than theoretical. We already provide one scaling result in Fig. 5, which shows that increasing the number of building tasks $N$ consistently improves the success rate of both AutoRPA and AutoRPA (code only), indicating that the synthesized RPA becomes more robust as trajectory diversity increases.
>
> We also conducted an additional complexity-based analysis on AndroidWorld (GPT-5). Grouping tasks by step complexity, AutoRPA achieves
> - 81.2% for tasks with step complexity ≤10 (48 tasks);
> - 78.7% for tasks with 10 < step ≤ 30 (47 tasks);
> - 57.1% for tasks with 30 < step < 120 (21 tasks).
> This confirms the reviewer’s intuition that long-horizon tasks remain more challenging, but also shows that the framework retains substantial potential beyond short-horizon skills. We will add this analysis and a more explicit discussion of failure modes in the revision.
>
> **Response to the key question**
>
> In our current implementation, the reusable RPA library is organized by task type, with one parameterized RPA function per task type. Therefore, as stated in Section 3.4, at test time, the system first checks whether a verified function exists for the current task type, then fills in parameters and executes it. In this setup, function lookup itself is lightweight; the dominant cost is inside the selected function (e.g., `find_element` or `ask_mllm`), rather than library retrieval. Regarding maintenance cost, the modular design keeps functions largely independent, so updates are local to the affected task type rather than global. For much larger libraries, we agree that building across task types and periodic re-verification would become increasingly important, and we see this as a promising direction for future work.

---

> > ### Author Rebuttal · Reviewer_ZCz8 · 2026-04-01
> >
> > Thank you for the detailed rebuttal and clarifications.
> > My questions have been largely addressed.
> >
> > However, concerns regarding the overall contribution and empirical limitations remain, and I am inclined to maintain my original score.

---

### Official Review · Reviewer_ZXqX · 2026-03-12

**Soundness:** 3
**Presentation:** 3
**Significance:** 3
**Originality:** 3
**Overall Recommendation:** 4
**Confidence:** 4

**Summary:**

Authors propose AutoRPA, a a framework that automatically distills the decision logic
of ReAct-style agents into robust RPA functions. It works in three phases: an Exploration Phase where a ReAct agent collects trajectories and a translator agent converts hard-coded actions (e.g., click(index=2)) into soft-coded, attribute-based equivalents. This is followed by an RPA Generation Phase where a builder agent synthesizes reusable RPA functions using tree-structured RAG over past trajectories> Finally, a Verification Phase employing a hybrid repair strategy which when generated code fails, a ReAct agent resumes from the breakpoint to produce a corrective demonstration for refinement. AutoRPA with GPT-5 planner outperforms the ReAct baseline on AndroidWorld, MiniWoB++.

**Compliance With Llm Reviewing Policy:**

Affirmed.

**Final Justification:**

Thank you for the detailed rebuttal and clarifications. I acknowledge the rebuttal from the authors but my concerns are still valid so I will keep my original score

**Key Questions For Authors:**

Mentioned above

**Limitations:**

Mentioned above

I think the proposed approach is interesting. However, it is unclear how scalable and useful it is compared to vanilla react baseline. The gains are marginal but the idea proposed is interesting and experiments are thorough.

**Strengths And Weaknesses:**

**Strengths**

- AutoRPA converts low-level actions to reusable programs (RPAs) that can be thought of as high-level reusable skills for complex interactions. This saves repititive reasoning required to execute same skills at test time.
- AutoRPA abstraction also allows to transfer the skills across platforms by modifying implementations of RPA to be platform specific. This can enable transfer of learned skills across platforms easily and is interpretable.
- AutoRPA also enables learning useful skills from failed trajectories too as it doesn’t depend on task suuccess rate to build useful RPA library.
- Experiment results are thorough and ablations clearly validate each component's contribution.

**Weaknesses**

- If I understand correctly the RPA at execution time still require DOM/accessibility tree to be able to execute the actions which makes it less flexible when compared to directly executing low-level action primitives using visual input.
- Building library of RPA seems to have a non-trivial upfront token cost (amortized after ~4 task executions). Analysis in figure 4 present token usage at inference time but it would also be great to see analysis on what is the token cost to build the RPA library and how well it scales. Does spending a lot more tokens during phase 1 enable building a much better RPA library or it saturates after a few demonstrations?
- The proposed method is still useful for building short horizon RPA skills. It is unclear how impactful it is to increase efficiency of the execution vs error rate. The improvement over simple ReAct baseline is quite small ~2-4%.  It is unclear whether this method is scalable to build medium to long horizon skills that are reusable and increase execution efficiency.

---

> ### Author Rebuttal · Authors · 2026-03-30
>
> **(1) Reliance on DOM / a11y trees**
>
> We agree that our default implementation prefers DOM/a11y trees when they are available, because structured UI metadata substantially improves robustness and lowers runtime cost. However, this is a pragmatic design choice rather than a hard requirement of the AutoRPA framework itself. We already discussed this limitation in Appendix A and noted that screenshot parsing tools such as OmniParser-V2 can alleviate it.
>
> To directly address the reviewer’s concern, we extend experiments in Table 5 and additionally evaluated a fully visual version of AutoRPA without any DOM/a11y trees on AndroidWorld using Claude-sonnet-4.5. In this setting, we use a screenshot-only ReAct Agent; the `find_element` and `ask_mllm` functions in both Translator and Builder rely directly on the grounder model for visual parsing and localization. To obtain the screenshot with SoM for better localization, we use the UI detector in OmniParser-V2  to outline the UI elements.
>
> | Method | Setting | Test Tokens (k) | Test SR (%) |
> | :--- | :--- | :--- | :--- |
> | ReAct$^\dagger$ | Screenshot+A11y tree | 146.0 | 76.7 |
> | ReAct$^\dagger$ | Screenshot Only | 145.1 | 69.0 |
> | ReAct$^\dagger$ | Screenshot Only (seed-2.0-pro) | 105.0 | 63.3 |
> | AutoRPA | Screenshot+A11y tree for All Agent | 36.1 | 76.9 |
> | AutoRPA | Screenshot Only for ReAct Agent | 33.5 | 70.1 |
> | AutoRPA | Screenshot Only for All Agent | 65.6 | 69.8 |
> | AutoRPA | Screenshot Only for All Agent, seed-2.0-pro as grounder | 60.5 | 69.3 |
>
> These results show that removing DOM/a11y indeed increases cost and slightly reduces performance, but AutoRPA remains effective and still uses substantially fewer test-time tokens than screenshot-only ReAct. Furthermore, we can introduce a much cheaper grounder model (e.g., seed-2.0-pro, priced at 1/13 of Claude-sonnet-4.5) during testing while maintaining a comparable success rate. This further supports our claim: ReAct spends most of its test-time budget on repeated "reasoning" rather than "grounding", and AutoRPA distills this reasoning logic into RPA code. We will include these experiments and discussions in the final version.
>
> **(2) Building costs and scaling**
>
> We agree that the build-time cost should be made more prominent. We already reported this analysis in Appendix C.1 (Table 4). The one-time build cost per task type is:
>
> - 233k tokens on AndroidWorld
> - 69k tokens on MiniWoB++ hard tasks
> - 42k tokens on MiniWoB++ overall
>
> Importantly, the dominant cost comes from the ReAct exploration stage, while translation, building, and verification are substantially smaller. We also already analyzed the effect of increasing the number of building tasks in Fig. 5: more demonstrations do improve robustness, but the gain shows clear diminishing returns, with performance beginning to saturate around 4–5 building tasks. Therefore, spending more phase-1 tokens helps initially, but it does not improve the library proportionally after a few demonstrations. We will make this build-cost analysis clearer.
>
> **(3) Practical impact and longer-horizon tasks**
>
> We respectfully emphasize that the main contribution of AutoRPA is efficiency without sacrificing success rate, rather than a large absolute SR gain over ReAct. For example, on AndroidWorld with GPT-5, ReAct achieves 74.1% SR with 142.5k test tokens, while AutoRPA achieves 75.9% SR with only 30.6k tokens; AutoRPA (code only) still achieves 70.7% SR with 6.2k tokens. We believe the 2–4% improvement should be interpreted together with the 82%–96% reduction in test-time token usage, which is exactly the objective of the paper.
>
> Regarding horizon length, we agree that very long-horizon reusable skills remain challenging, and this should be stated more explicitly. To better quantify this, we additionally bucketed AndroidWorld tasks by complexity and found that AutoRPA (GPT-5) achieves
> - 81.2% for tasks with step complexity ≤10 (48 tasks);
> - 78.7% for tasks with 10 < step ≤ 30 (47 tasks);
> - 57.1% for tasks with 30 < step < 120 (21 tasks).
>
> This suggests that the method already extends beyond very short tasks, but performance does degrade on longer-horizon tasks. We will clarify this boundary in the revision.

---

### Official Review · Reviewer_EkHH · 2026-03-13

**Soundness:** 2
**Presentation:** 3
**Significance:** 3
**Originality:** 2
**Overall Recommendation:** 4
**Confidence:** 3

**Summary:**

AutoRPA is a framework for automatically generating Robotic Process Automation (RPA) scripts for repetitive GUI tasks. It distills the step-by-step reasoning of ReAct-style LLM agents into reusable Python code through exploration, RPA generation with RAG, and verification with hybrid repair. Evaluated on diverse GUI benchmarks, AutoRPA matches or exceeds the success rates of baseline agents while achieving drastic reductions in token consumption and computational overhead, significantly enhancing the runtime efficiency of repetitive automation tasks.

**Compliance With Llm Reviewing Policy:**

Affirmed.

**Key Questions For Authors:**

1. The experiments default to using $N=3$ tasks for each task type during the building stage. How sensitive is the generated RPA code to the diversity of these building tasks? If the $N$ tasks sampled are highly similar or lack edge cases, does the Builder agent tend to overfit the synthesized RPA script? Are there automated strategies for selecting a diverse set of instances to ensure maximum coverage during the building phase?
2. AutoRPA addresses minor layout changes by translating hard-coded actions into soft-coded equivalents using semantic attributes like text content or element types. However, how does the framework handle significant UI redesigns where the semantic attributes themselves change (e.g., a button labeled "Submit" is replaced by an unlabeled "Arrow" icon)? Does the system automatically detect this out-of-distribution failure and autonomously trigger a new Exploration Phase to rebuild the RPA, or does it simply fail and wait for user intervention?

**Limitations:**

yes

**Strengths And Weaknesses:**

1. Strengths
- The proposed framework offers a novel agent workflow that efficiently distills complex ReAct trajectories into deterministic, adaptive RPA scripts. This work addresses a critical practical bottleneck by drastically reducing the prohibitive inference costs typically associated with LLM agents in repetitive GUI automation.
- The methodology is supported by rigorous evaluation across three diverse benchmarks (AndroidWorld, WebArena, and MiniWoB++). Furthermore, ablation studies validate the essential contributions of the Translator, RAG mechanism, and hybrid repair strategy.
- The manuscript is well-organized and logically structured, with Figures 2 and 3 providing intuitive visual clarity for the code generation and refinement pipeline.
2. Weaknesses
- The framework's reliance on DOM or accessibility trees limits its applicability in purely visual environments without advanced parsing support.
- The Builder agent occasionally generates redundant logic or over-relies on retrieval tools rather than direct MLLM querying, presenting a potential bottleneck for highly complex, long-horizon tasks.

---

> ### Author Rebuttal · Authors · 2026-03-29
>
> **(1) Reliance on DOM / a11y trees**
>
> We agree that relying on DOM or a11y trees has limitations in certain extreme scenarios. First, it is important to note that DOM or a11y trees are inherently accessible in most mainstream GUI systems (e.g., Web, Android, iOS, UIA API for Windows, Accessibility API for macOS, AT-SPI for Linux). As discussed in our 'Limitations', relying on them does improve the stability and execution efficiency of RPA scripts, but this can be mitigated using visual parsing tools like OmniParser-V2. In Table 5, we found that using a screenshot-only ReAct Agent decreased the success rate (76.9% to 70.1%).
>
> To directly address the reviewer’s concern, we extend experiments in Table 5 and additionally evaluated a fully visual version of AutoRPA without any DOM/a11y trees on AndroidWorld using Claude-sonnet-4.5. In this setting, we use a screenshot-only ReAct Agent; the `find_element` and `ask_mllm` functions in both Translator and Builder rely directly on the grounder model for visual parsing and localization. To obtain the screenshot with SoM for better localization, we use the UI detector in OmniParser-V2  to outline the UI elements.
>
> | Method | Setting | Test Tokens (k) | Test SR (%) |
> | :--- | :--- | :--- | :--- |
> | ReAct$^\dagger$ | Screenshot+A11y tree | 146.0 | 76.7 |
> | ReAct$^\dagger$ | Screenshot Only | 145.1 | 69.0 |
> | ReAct$^\dagger$ | Screenshot Only (seed-2.0-pro) | 105.0 | 63.3 |
> | AutoRPA | Screenshot+A11y tree for All Agent | 36.1 | 76.9 |
> | AutoRPA | Screenshot Only for ReAct Agent | 33.5 | 70.1 |
> | AutoRPA | Screenshot Only for All Agent | 65.6 | 69.8 |
> | AutoRPA | Screenshot Only for All Agent, seed-2.0-pro as grounder | 60.5 | 69.3 |
>
> These results show that removing DOM/a11y indeed increases cost and slightly reduces performance, but AutoRPA remains effective and still uses substantially fewer test-time tokens than screenshot-only ReAct. Furthermore, we can introduce a much cheaper grounder model (e.g., seed-2.0-pro, priced at 1/13 of Claude-sonnet-4.5) during testing while maintaining a comparable success rate. This further supports our claim: ReAct spends most of its test-time budget on repeated "reasoning" rather than "grounding", and AutoRPA distills this reasoning logic into RPA code. We will include these experiments and discussions in the final version.
>
> **(2) Builder redundancy**
>
> Our interpretation of the reviewer’s concern is that the Builder may sometimes generate unnecessarily complicated logic or rely too much on trajectory retrieval, instead of using simpler MLLM reasoning (e.g., ask_mllm) when appropriate. First, our initial design goal was to reduce token consumption during testing, so the RPA code generated by the Builder only uses ask_mllm when necessary. Second, we would like to emphasize that retrieval is introduced for a concrete reason: when the Builder only sees simplified trajectories, it can make incorrect assumptions about the actual UI state, which motivates our RAG design. Empirically, this component is helpful but not dominant: in Table 3, removing RAG reduces success from 51.7% to 48.8%, rather than causing a collapse. In our design, to limit the overhead, we enforce a strict upper limit of 3 RAG calls for the Builder.
>
> **(3) Sensitivity to building tasks**
>
> This is a very insightful question. As demonstrated in Figure 5, increasing the number of building tasks (N) indeed helps improve the success rate during testing. We completely agree that if the sampled building tasks are highly similar or lack edge cases, the Builder tends to overfit and generate RPA scripts lacking generalization. Regarding automated strategies to select diverse instances for maximum coverage, there are currently some LLM-based "curiosity-driven" automatic exploration methods used for building agent datasets. However, considering the execution complexity and unreliability of such methods in long-horizon tasks, we have not integrated this mechanism in the current framework to preserve its conciseness. This is indeed a highly valuable direction for future research, and we will add it to the Future Work section.
>
> **(4) Handling major UI redesigns**
>
> Thank you for pointing out this critical challenge in real-world applications. First, for minor UI changes or routine matching failures, AutoRPA automatically triggers the visual grounder as a fallback strategy when attribute matching is insufficient. For significant UI redesigns, the current code-only setting may fail. In the paper’s evaluation protocol, we do not automatically launch a fresh Exploration/Building loop at test time, because this would make comparison against static baselines unfair. We agree that an always-online Building version that detects repeated failures and then re-enters the Exploration/Verification loop is a practical and natural extension of our framework. We will add a detailed discussion on this "continuous learning and rebuilding" mechanism in the revision.

---

> > ### Author Rebuttal · Reviewer_EkHH · 2026-04-04
> >
> > I believe the authors have adequately addressed my comments/questions.

---

### Decision · Program_Chairs · 2026-04-30

**Decision:**

Accept (regular)

**Comment:**

This paper proposes AutoRPA, a framework that distills ReAct-style LLM agent trajectories into reusable RPA functions for repetitive GUI automation. The system uses a translator-builder-verifier pipeline with retrieval-augmented code synthesis and a hybrid repair strategy. Experiments on AndroidWorld, WebArena, and MiniWoB++ demonstrate 82-96% token reduction while maintaining competitive task success rates.
All three reviewers agree the paper addresses a practical and well-motivated problem — the inefficiency of repeated LLM reasoning for recurring GUI tasks. The translator-builder-verifier pipeline is well-structured and clearly presented (EkHH, ZCz8). The efficiency gains are substantial and consistent across benchmarks and backbone models (EkHH, ZXqX, ZCz8).

Overall, the paper makes a solid, practical contribution to GUI automation. The evaluation is thorough across three benchmarks with multiple backbones. The paper would benefit from more prominent discussion of build-time costs and clearer characterization of the complexity boundary where the approach degrades.